# SR-Scientist: Scientific Equation Discovery With Agentic AI

**Shijie Xia**[1,2,3*] **Yuhan Sun**[1,3*] **Pengfei Liu**[1,2,3‡]
[1]Shanghai Jiao Tong University [2]Shanghai Innovation Institute [3]GAIR
{shijiexia,sunmouren,pengfei}@sjtu.edu.cn

## Abstract

Recently, Large Language Models (LLMs) have been applied to scientific equation discovery, leveraging their embedded scientific knowledge for hypothesis generation. However, current methods typically confine LLMs to the role of an equation proposer within search algorithms like genetic programming. In this paper, we present SR-Scientist, a framework that elevates the LLM from a simple equation proposer to an autonomous AI scientist that writes code to analyze data, implements the equation as code, submits it for evaluation, and optimizes the equation based on experimental feedback. Specifically, we wrap the code interpreter into a set of tools for data analysis and equation evaluation. The agent is instructed to optimize the equation by utilizing these tools over a long horizon with minimal human-defined pipelines. Empirical results show that SR-Scientist outperforms baseline methods by an absolute margin of 6% to 35% on datasets covering four science disciplines. Additionally, we demonstrate our method's robustness to noise, the generalization of the discovered equations to out-of-domain data, and their symbolic accuracy. Furthermore, we develop an end-to-end reinforcement learning framework to enhance the agent's capabilities[1].

## 1 Introduction

In the era of Agentic AI, Large Language Models (LLMs) have evolved from simple knowledge retrievers to agentic models capable of completing complex tasks by interacting with their environments (Schneider, 2025), as exemplified by products like Claude Code (Anthropic, 2025) and Gemini CLI (Google, 2025). These agents exhibit many characteristics of human scientists, such as engaging in long-horizon interaction with environmental feedback, operating with autonomy, and relying less on predefined pipelines (Newell et al., 1972). However, in most current work using LLMs for scientific discovery (Shojaee et al., 2025a; Novikov et al., 2025; Romera-Paredes et al., 2024), LLMs serve as static components within human-crafted pipelines, lacking the autonomy to generate and refine hypotheses through active environmental interaction. Building a scientific discovery framework around agentic models could therefore shift the paradigm from using LLMs as passive tools to empowering them as autonomous agents that drive the entire discovery lifecycle. In this paper, we focus on equation discovery, a fundamental task in science.

Mathematical equations are central to scientific progress, serving as concise and interpretable models of physical phenomena (Lemos et al., 2022; Batra et al., 2021; Hernandez et al., 2019). The task of data-driven equation discovery, also known as symbolic regression (SR), is an NP-hard problem due to its vast search space (Virgolin & Pissis, 2022). Traditional methods have relied on Genetic Programming (GP) for combinatorial search (Cranmer, 2023) or on deep neural networks trained on large-scale synthetic data for direct prediction (Kamienny et al., 2022; Biggio et al., 2021), which often suffer from efficiency and scalability issues. Recently, the research community has begun to embed LLMs into GP algorithms as equation proposers, leveraging their scientific prior knowledge to generate more effective hypotheses (Shojaee et al., 2025a; Grayeli et al., 2024). While this can enhance search efficiency, LLMs primarily serve as equation generators in a fixed pipeline, which lacks the autonomy to gain insight for equation design through tools.

---

[*]Co-first authors. [‡]Corresponding author.
[1]Code and data are available: https://github.com/GAIR-NLP/SR-Scientist

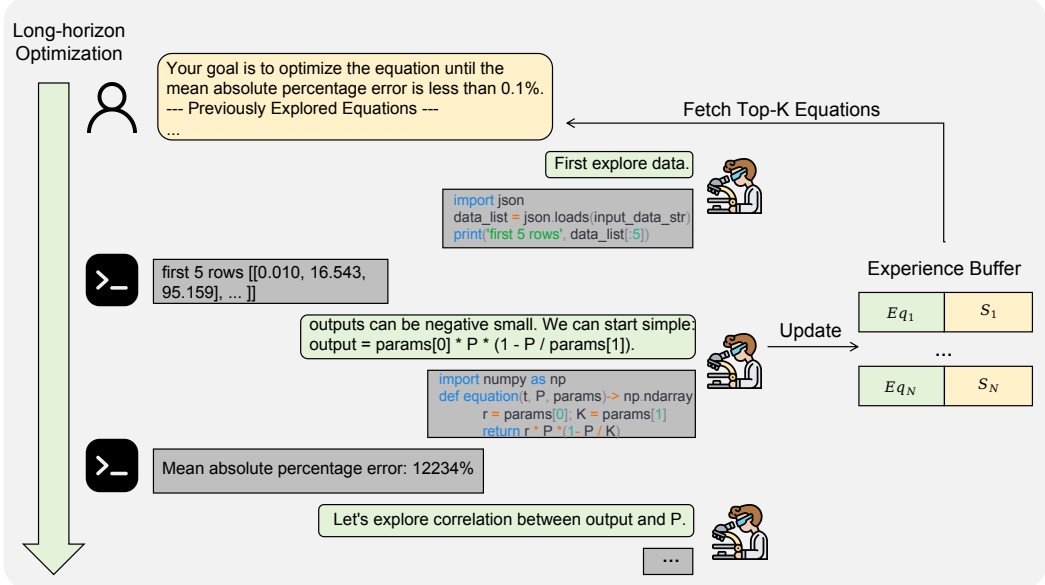

Figure 1: The inference framework of SR-SCIENTIST. At each iteration, the LLM agent autonomously conducts long-horizon optimization using code interpreters for data analysis and equation evaluation. To overcome the context length limitations of current LLMs, we implement an experience buffer to fetch the best-performing equations for subsequent iterations. 'Eq' denotes the equation and 'S' denotes the equation score.

To address these limitations, we introduce SR-SCIENTIST, a framework designed to enable LLMs to find scientific equations through long-horizon optimization driven by experimental feedback and analysis. To achieve this, we wrap the code interpreter as tools to analyze data and evaluate equations. As shown in Figure 1, the LLM agent is instructed to solve problems by utilizing these tools, including writing code to analyze data, implementing the equation as code, submitting it for evaluation, and optimizing the equation based on experimental feedback. In the process, we emphasize long-horizon optimization, allowing the agent to interact with data through tools over multiple turns (e.g., more than twenty) to gather sufficient information for equation design. To overcome the context length limitations of current LLMs, we implement an experience buffer to store the explored equations and fetch the best-performing equations for subsequent iterations. Moreover, we adhere to the principle of minimal human-defined pipelines (Sutton, 2019), in which the agent is free to determine its own workflow for a given problem. Within the flexible framework, we develop a reinforcement learning (RL) pipeline from training data construction to reward design to help the LLM agent evolve its capabilities.

To validate the effectiveness of our framework, we evaluate SR-SCIENTIST using 5 different models as backbone LLMs on datasets covering 4 science disciplines carefully designed to prevent the LLMs from relying on memorized equations from their training data (Shojaee et al., 2025b). Results demonstrate that SR-SCIENTIST consistently outperforms state-of-the-art SR methods, with four of the models achieving an absolute margin of 6% to 35% over baselines. We also observe significant improvement after applying RL training. Additionally, we demonstrate our method's robustness to noise, the generalization of the discovered equations to out-of-domain data, and their symbolic accuracy. Furthermore, our ablation studies highlight the importance of data analysis, the experience buffer, and long-horizon optimization.

Overall, our contributions are as follows:

- We develop SR-SCIENTIST, a framework where an autonomous agent discovers scientific equations through long-horizon, tool-driven data analysis and equation evaluation.
- We show that SR-SCIENTIST significantly outperforms baseline methods in precision, generalization, robustness to noise, and symbolic accuracy.

- We develop a corresponding end-to-end reinforcement learning pipeline to enhance the agent's capabilities.

## 2 RELATED WORK

**Symbolic Regression**  SR aims to identify an interpretable mathematical expression that accurately describes the relationship between variables in observed data (Makke & Chawla, 2024; Cava et al., 2021). Current popular methods can be categorized as follows: 1) *GP-based Methods*: This line of methods uses constrained representations, such as expression trees, to define a search space composed of symbolic operators, constants, and variables (Cranmer, 2023). Evolutionary or genetic programming is then applied to search for solutions. 2) *Deep learning Methods*: These methods leverage deep neural networks to learn a direct mapping from numerical observations to their underlying mathematical expressions through online (Petersen et al., 2021; Landajuela et al., 2022) or offline learning (Kamienny et al., 2022; Biggio et al., 2021). 3) *LLM-based methods*: Due to the efficiency and scalability issues of the above methods, the research community has integrated LLMs with traditional techniques like GP, leveraging the models' extensive knowledge to generate more informed hypotheses (Shojaee et al., 2025a; Grayeli et al., 2024; Ma et al., 2024; Wang et al., 2025b). While promising, current hybrid methods still lack the direct analysis of observed data through tools to gain insights. Furthermore, the LLM's behavior is often fixed, lacking the autonomy to decide its actions in a goal-oriented manner. Moreover, most work focuses on using LLMs for inference, without exploring how these models evolve their abilities through methods like RL.

**Agentic AI**  Agentic AI can autonomously execute complex tasks in dynamic environments requiring adaptation, interaction, and reasoning in a goal-oriented manner (Schneider, 2025). The open-source community has built powerful agentic models for tasks like software engineering (Team et al., 2025; Zeng et al., 2025a), search (Gao et al., 2025), and GUI operations (Qin et al., 2025; Wang et al., 2025a), enabling them to solve tasks in an end-to-end manner with minimal human intervention. However, there has been limited focus on scientific discovery. In this paper, we demonstrate that with careful design, these agentic models can be powerful for equation discovery and can also improve their abilities through RL.

## 3 SR-SCIENTIST

### 3.1 PROBLEM FORMULATION

Symbolic regression aims to find a symbolic expression that describes the underlying relationship in a dataset. Formally, given a dataset $\mathcal{D} = \{(\mathbf{x}_i, y_i)\}_{i=1}^n$, where $\mathbf{x}_i \in \mathbb{R}^d$ is an input vector and $y_i \in \mathbb{R}$ is a scalar output, an SR method seeks to find a concise symbolic function $f$ such that $f(\mathbf{x}_i) \approx y_i$. The goal of SR is not only to discover a function that minimizes the error between the predicted values and the true values but also one that generalizes well to unseen inputs while maintaining interpretability.

### 3.2 INFERENCE FRAMEWORK

We outline the overall inference framework in Algorithm 1. At each iteration, we set a goal $G_i$ for a desired optimization precision and the agent is instructed to find an equation that satisfies the precision. We choose the mean absolute percentage error[2] (MAPE) as the goal: MAPE $= \frac{100\%}{n} \sum_{i=1}^n \left| \frac{y_i - f(\mathbf{x}_i)}{y_i} \right|$. Then, the LLM agent solves the problem by interleaving internal reasoning with external environment feedback. Formally, the agent's trajectory is structured as follows, typically within a ReAct (Yao et al., 2023) framework:

$$(r_1, \mathcal{T}_1, o_1), (r_2, \mathcal{T}_2, o_2), ..., (r_k, \mathcal{T}_k, o_k) \tag{1}$$

---

[2]This is slightly different from other SR methods that usually use the mean squared error (MSE) or normalized mean squared error (NMSE) as the score, as we find MAPE provides a more uniform precision target across different data than the aforementioned metrics. To ensure a fair comparison with other SR methods, we select the equation with the lowest NMSE in practical evaluation.

where $r_i$ denotes the model's natural language reasoning at step $i$, $\mathcal{T}_i \subseteq \mathcal{T}$ is the set of tools invoked at step $i$, and $o_i$ is the observation received after executing the tools in $\mathcal{T}_i$. We also set a maximum number of interaction turns $M$ for the trajectory to avoid excessively long inference times without improved performance.

**Tool Design** We provide a code interpreter as the primary tool, enabling the agent to interact with the data and validate hypotheses. Specifically, we wrap the code interpreter into two common tools for the tasks, `data analyzer` and `equation evaluator`, denoted as $T_1$ and $T_2$, respectively. For $T_1$, we link it to the observed data and include a code example in the prompt demonstrating how to access the data. Through this tool, the agent can write and execute diverse code to analyze the data, such as inspecting data samples, conducting statistical analysis, or analyzing the residuals between the predicted value and the true value[3]. Notably, we do not constrain the code to these examples, allowing for diverse and emergent analysis patterns. For $T_2$, following the approach of Shojaee et al. (2025a), we design the tool to accept an equation skeleton with placeholders for constants in code format (See Figure 1 or 4 for the equation example). During the execution phase, the BFGS algorithm is implemented to optimize these constants and then report the performance of the equation. It also accepts equations with constants decided through other methods like data analysis. This tool prevents the agent from writing repetitive code for equation evaluation during the long-horizon exploration process.

**Experience Buffer** While the agent can continue the optimization process over multiple turns, the model's limited context length presents a challenge. Moreover, the equations can still perform poorly after a long-horizon optimization. To overcome this, we maintain an experience buffer $E = \{(e_i, s_i)\}_{i=1}^{N}$ to record the equations the agent has explored, where $e_i$ denotes the equation and $s_i$ is its corresponding MAPE. At the beginning of each iteration, we fetch the best $K$ equations from the buffer and provide them to the agent as in-context examples. The optimization goal is also updated if it was reached in the previous iteration. This mechanism effectively bypasses the context length limitation. Additionally, we explored using GP algorithms for experience management but observed no significant improvement.

**Stopping Condition and Submission** We stop the exploration process when the maximum number of iterations is reached or the equation's error is sufficiently small. Then, we select the equation with the best performance on the observed data and submit it for final evaluation.

---

**Algorithm 1** SR-SCIENTIST inference framework.

**Input:** Iterations N; maximum turns $M$; number of fetched equations $K$; the initial goal $G_1$

```
1:  # Store ranked equations by heap
2:  H ← heap()
3:  for i = 1, . . . , N do
4:      # Generate candidate equations
        from LLM
5:      if i == 1 then
6:          E_i ← LLM(M, G_i)
7:      else
8:          E_i ← LLM(H.topk(K), M, G_i)
9:      end if
10:     H.append(E_i)
11:     # Check stopping condition and
        update goal
12:     if stopping_condition(H) then
13:         break
14:     else
15:         G_{i+1} ← UpdateGoal(H, G_i)
16:     end if
17: end for
```

**Output:** `H.topk(1) # Return the best`

---

### 3.3 TRAINING FRAMEWORK

**Training Data Construction** Following the approach of Shojaee et al. (2025b), we employ a mixed strategy of rule-based and model-based data synthesis. For each scientific scenario covering multiple variables, we instruct an LLM to synthesize potential relationships between the variables, typically in the form of an equation skeleton. For each skeleton, we use the model to determine the values for the constants based on their physical significance and thereby construct the full equation.

---

[3]Since we primarily use a text-only model that cannot accept figures as input, we explicitly instruct the model via prompts to avoid generating plots. However, the model backbone is orthogonal to our framework, which intrinsically supports such plot analysis patterns.

Once the complete equation is formulated, we define an appropriate range of values for its variables and synthesize the observed data accordingly. This dataset is then split into training and evaluation sets. The training data is accessible to the agent during rollouts, while the evaluation data is used to measure performance and calculate the reward.

**Reward Design** In the rollout process, the LLM agent is instructed to find an equation that satisfies the precision specified by a MAPE goal $s_{\text{goal}}$. Then, the LLM agent interacts with the environment and generates multiple equations. To simplify the RL infrastructure, the agent conducts the optimization process for one iteration. Unlike math or code tasks that typically assign binary rewards based on the outcome (DeepSeek-AI et al., 2025; Zeng et al., 2025b), the performance of equations can be measured by continuous metrics such as MAPE. This makes it possible to assign continuous rewards to avoid reward sparsity. Since we only submit the best equation at inference (Petersen et al., 2021), we select the best equation from those explored and use its score $s$ for the calculation. We employ a log-linear reward function that maps $s$ to the range $[0, 1]$ as follows:

$$\mathcal{R} = \text{clip}(\frac{\lg s_{\max} - \lg s}{\lg s_{\max} - \lg s_{\text{goal}}}, 0, 1) \tag{2}$$

where $s_{\max}$ represents the maximum MAPE for which a non-zero reward can be gained. Additionally, we explore other reward functions and present the discussion in Appendix C.2.

**Training Algorithm** We apply the Group Relative Policy Optimization (GRPO) (Shao et al., 2024) algorithm for optimization. To encourage exploration, we omit the KL penalty term against a reference model and observe that this leads to faster convergence and comparable performance. Specifically, for each question $q$, we sample a group of outputs $\{o_1, o_2, \cdots, o_G\}$ from the old policy $\pi_{\theta_{\text{old}}}$ and then optimize the policy $\pi_\theta$ by maximizing the following objective:

$$\mathcal{J}_{GRPO}(\theta) = \mathbb{E}[q \sim P(Q), \{o_i\}_{i=1}^G \sim \pi_{\theta_{old}}(O|q)]$$
$$\frac{1}{G} \sum_{i=1}^G \left( \min \left( \frac{\pi_\theta(o_i|q)}{\pi_{\theta_{old}}(o_i|q)} A_i, \text{clip} \left( \frac{\pi_\theta(o_i|q)}{\pi_{\theta_{old}}(o_i|q)}, 1 - \epsilon, 1 + \epsilon \right) A_i \right) \right) \tag{3}$$

where $\varepsilon$ is the hyper-parameter, and $A_i$ is the advantage computed using a group of rewards corresponding to the outputs within each group.

## 4 EXPERIMENTS

### 4.1 SETUP

**Dataset** Due to the vast corpus that LLMs have been pretrained on, the evaluation dataset must be carefully curated to prevent LLMs from having memorized the equations. We evaluate our methods on the synthetic part of the LLM-SRBench (Shojaee et al., 2025b), denoted as LSR-Synth[4]. It combines known terms in the underlying equation with synthetic, novel terms to create problems that go beyond memorization. Furthermore, all equations and visualizations of their generated data points were verified by two subject matter experts to ensure scientific rigor. It contains 129 problems, spanning four scientific domains: chemistry (36), biology (24), physics (44), and material science (25). For each problem, it has three types of datasets: a training set accessible to the SR method, an in-domain (ID) test set, and an out-of-domain (OOD) test set.

**Evaluation Metrics** We use accuracy-to-tolerance as our main metric. We find it serves as a more robust metric when aggregating the overall results covering multiple problems than others, like Normalized Mean Squared Error. It is also more challenging than other metrics like $R^2$. Please refer to Appendix D.1 for a detailed analysis. Given a desired tolerance threshold $\tau$, the metric calculates whether the predicted values and the ground truth values satisfy it as follows:

$$\text{Acc}_\tau = \mathbb{1} \left( \max_{1 \leq i \leq N_{\text{test}}} \left| \frac{\hat{y}_i - y_i}{y_i} \right| \leq \tau \right) \tag{4}$$

---

[4]We also considered LSR-Transform (Shojaee et al., 2025b) as a candidate for the evaluation dataset. However, we found that it might still have memorization issues, as nearly 50% of the problems achieved a sufficiently small error after the first iteration.

| Method | Overall | | Material Science | | Chemistry | | Biology | | Physics | |
|---|---|---|---|---|---|---|---|---|---|---|
| | $Acc_{0.01}$ | $Acc_{0.001}$ | $Acc_{0.01}$ | $Acc_{0.001}$ | $Acc_{0.01}$ | $Acc_{0.001}$ | $Acc_{0.01}$ | $Acc_{0.001}$ | $Acc_{0.01}$ | $Acc_{0.001}$ |
| *Without LLMs* | | | | | | | | | | |
| GPLearn | 0.00 | 0.00 | 0.00 | 0.00 | 0.00 | 0.00 | 0.00 | 0.00 | 0.00 | 0.00 |
| E2E | 0.26 | 0.00 | 1.33 | 0.00 | 0.00 | 0.00 | 0.00 | 0.00 | 0.00 | 0.00 |
| NeSymReS | 3.10 | 0.78 | 8.00 | 4.00 | 2.78 | 0.00 | 4.17 | 0.00 | 0.00 | 0.00 |
| DSR | 0.00 | 0.00 | 0.00 | 0.00 | 0.00 | 0.00 | 0.00 | 0.00 | 0.00 | 0.00 |
| uDSR | 29.46 | 12.40 | 36.00 | 8.00 | 50.00 | 25.00 | 29.17 | 8.33 | 9.09 | 6.82 |
| PySR | 29.46 | 14.47 | 53.33 | 22.67 | 25.93 | 11.11 | 16.67 | 6.95 | 25.76 | 16.67 |
| *Qwen3-Coder-480B-A35B-Instruct* | | | | | | | | | | |
| LaSR | 11.89 | 7.49 | 13.33 | 6.67 | 16.67 | 9.26 | 6.95 | 6.95 | 9.85 | 6.82 |
| LLM-SR | 41.08 | 18.09 | 80.00 | 52.00 | 36.11 | 9.26 | 30.56 | 18.06 | 28.79 | 6.06 |
| SR-SCIENTIST | 49.09 | 24.55 | **86.67** | 69.33 | 40.74 | 5.56 | 50.00 | 26.39 | 34.09 | 13.64 |
| *GLM-4.5-Air* | | | | | | | | | | |
| LaSR | 14.21 | 8.53 | 18.67 | 12.00 | 16.67 | 10.18 | 11.11 | 5.56 | 11.36 | 6.82 |
| LLM-SR | 35.92 | 14.47 | 61.33 | 38.67 | 30.56 | 3.71 | 18.06 | 4.17 | 35.61 | 15.15 |
| SR-SCIENTIST | 48.32 | 25.06 | 81.33 | **70.67** | 45.37 | 11.11 | 40.28 | 16.66 | 36.37 | 15.15 |
| *GPT-OSS-120B* | | | | | | | | | | |
| LaSR | 16.02 | 10.08 | 20.00 | 12.00 | 18.52 | 12.96 | 9.72 | 5.56 | 15.15 | 9.09 |
| LLM-SR | 28.16 | 11.37 | 64.00 | 38.67 | 22.22 | 3.71 | 11.11 | 2.78 | 21.97 | 6.82 |
| SR-SCIENTIST | **63.57** | **49.35** | 74.67 | 60.00 | **81.48** | **64.81** | **66.67** | **43.05** | **40.91** | **34.09** |
| *GPT-OSS-20B* | | | | | | | | | | |
| LaSR | 12.66 | 8.53 | 14.67 | 8.00 | 15.74 | 12.96 | 11.11 | 6.95 | 9.85 | 6.06 |
| LLM-SR | 33.33 | 12.92 | 70.67 | 46.67 | 28.71 | 5.56 | 22.22 | 2.78 | 21.97 | 5.30 |
| SR-SCIENTIST | 42.64 | 23.00 | 62.67 | 40.00 | 49.07 | 23.15 | 34.72 | 19.44 | 30.30 | 15.15 |
| *Qwen3-Coder-30B-A3B-Instruct* | | | | | | | | | | |
| LaSR | 12.66 | 8.27 | 20.00 | 10.67 | 17.59 | 12.04 | 8.33 | 6.95 | 6.82 | 4.55 |
| LLM-SR | 24.55 | 7.24 | 29.33 | 6.67 | 30.55 | 10.18 | 15.28 | 5.55 | 21.97 | 6.06 |
| SR-SCIENTIST | 32.30 | 16.02 | 81.33 | 52.00 | 22.22 | 5.56 | 22.22 | 8.33 | 18.18 | 8.33 |
| + RL | 40.92 | 20.69 | 85.33 | 65.33 | 37.38 | 7.46 | 29.17 | 11.11 | 25.00 | 11.37 |

Table 1: Comparison of SR-SCIENTIST and SR baseline models on different scientific benchmark problems measured by $Acc_{0.01}$ and $Acc_{0.001}$.

We discard the 5% worst predictions to prevent outliers due to the presence of the max operator, following Biggio et al. (2021); Kamienny et al. (2022). Additionally, we present symbolic accuracy, which checks if the discovered equations are identical to the ground truth equations. We present the details for the symbolic accuracy calculation in Appendix D.6.

**Baselines** We comprehensively compare our methods against several baseline methods. For methods without LLMs, we include GPLearn, E2E (Kamienny et al., 2022), NeSymReS (Biggio et al., 2021), DSR (Petersen et al., 2021), uDSR (Landajuela et al., 2022), and PySR (Cranmer, 2023). For the LLM-based methods, we include LLM-SR and LaSR for comparison. For methods without LLMs, we constrain the number of equation candidates for each problem to 100,000. For methods with LLMs, we constrain the LLM calls to 1,000, with at most 1,000 equation candidates from each problem. For SR-SCIENTIST, we set the maximum number of turns per iteration to 25, with a total of 40 iterations. We present the configuration details of the baseline methods in Appendix A and the configuration of SR-SCIENTIST in Appendix B. We evaluate our method with different models as backbone LLMs, including Qwen3-Coder-480B-A35B (Qwen, 2025), GLM-4.5-Air (Zeng et al., 2025a), GPT-OSS-120B (OpenAI, 2025), GPT-OSS-20B, and Qwen3-Coder-30B-A3B. Due to computation costs and training infrastructure limitations, we use only Qwen3-Coder-30B-A3B for RL training, while the other models are used for inference only. However, we also verify the effectiveness of RL training on other models and present the results in Appendix C.4. For each experiment, we repeat it three times and report the average value to reduce noise.

**Training Details** For training data construction, we synthesize training data for the scientific scenarios in LSR-Synth. To prevent data contamination, two authors of this paper independently and carefully checked the equation skeletons to prevent duplicates and confirmed their agreement. Finally, we obtained 1024 problems for RL training. Please refer to Appendix C.1 for details. During

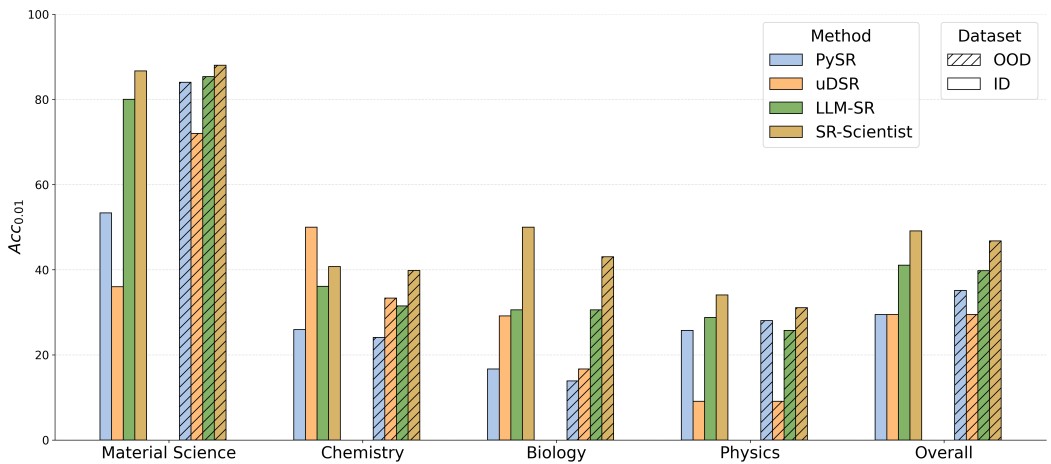

Figure 2: Detailed results of in-domain (ID) and out-of-domain (OOD) performance using $Acc_{0.01}$ across various LSR-Synth scientific domains. (with Qwen3-Coder-480B as LLM backbone)

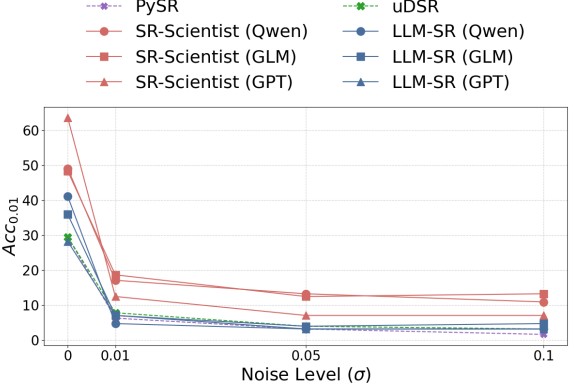

Figure 3: Noise robustness analysis on LSR-Synth. Qwen, GLM, and GPT represent Qwen3-Coder-480B, GLM-4.5-Air, and GPT-OSS-120B, respectively.

| Method | SA |
|---|---|
| uDSR | 0.77 |
| PySR | 4.65 |
| LLM-SR (GLM) | 5.43 |
| LLM-SR (GPT) | 4.65 |
| LLM-SR (Qwen) | 5.43 |
| SR-SCIENTIST (GLM) | 7.75 |
| SR-SCIENTIST (GPT) | 7.00 |
| SR-SCIENTIST (Qwen) | 7.00 |

Table 2: Symbolic accuracy (SA) of different methods on LSR-Synth. GLM, GPT, and Qwen represent GLM-4.5-Air, GPT-OSS-120B, and Qwen3-Coder-480B, respectively.

rollout, we set the maximum number of turns to 20 and train for 60 steps. We set $s_{max}$ to 100% and $s_{goal}$ to 0.1% in the reward design. The detailed training configure can be found in the Appendix C.3.

## 4.2 MAIN RESULTS

**Precision** Table 1 presents the accuracy results. In terms of overall accuracy, SR-SCIENTIST consistently outperforms the baseline methods, with four of the models achieving an absolute performance margin of 6% to 35% over the baselines. Notably, when using GPT-OSS-120B as a backbone, SR-SCIENTIST achieves the highest overall performance, with an $Acc_{0.01}$ of 63.57% and an $Acc_{0.001}$ of 49.35%. In terms of performance across different subjects, with backbones such as Qwen3-Coder-480B, GLM-4.5-Air, and GPT-OSS-120B, SR-SCIENTIST surpasses other LLM-based methods in all subjects. For Qwen3-Coder-30B, end-to-end RL training significantly improves its performance compared to the original model in all subjects, highlighting that the agent can evolve its abilities through its own experience. We provide further analysis of the LLM backbone for RL training in Appendix C.4 and the computation cost discussion in Appendix D.2.

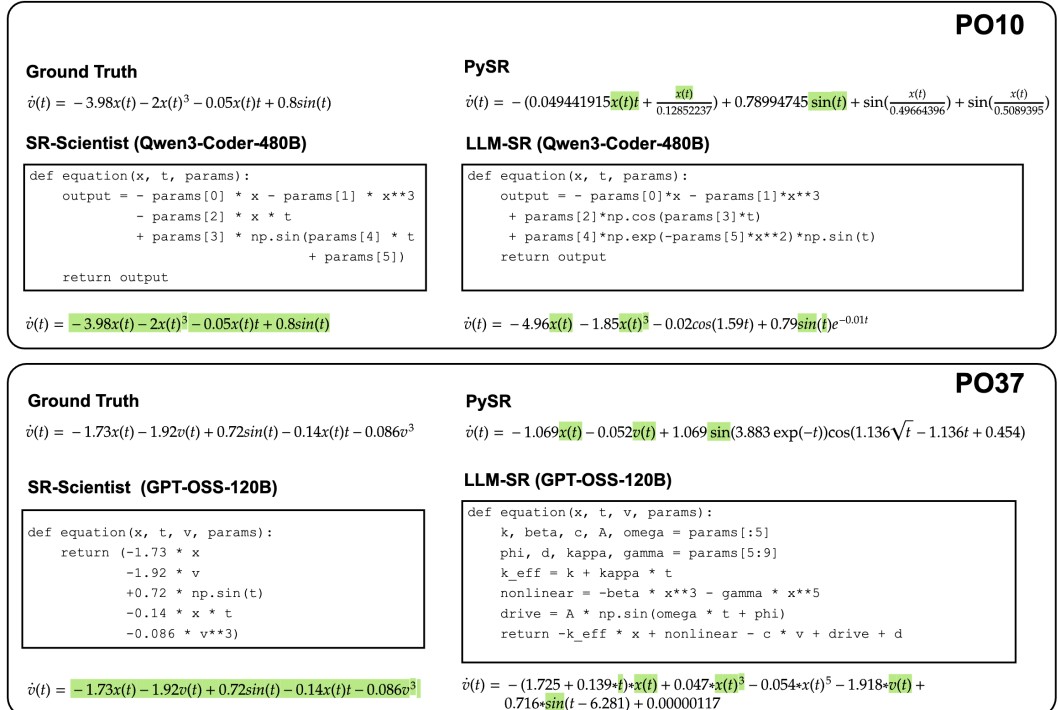

Figure 4: Equations discovered for the PO10 and PO37 physics problems. The variables $\dot{v}(t)$, $t$, $x$, and $v$ represent acceleration in a non-linear harmonic oscillator, time, position, and velocity, respectively. Terms highlighted in green are common to both the predicted and ground truth equations.

**OOD Performance**    Figure 2 shows the performance of the discovered equations on OOD data. For material science, the performance of all methods improves when shifting from ID to OOD data, while for other subjects, the methods exhibit varying trends. Among them, SR-SCIENTIST still achieves the best performance on OOD data, demonstrating its strong generalization capabilities. The results of other models are presented in Appendix D.3 and show similar trends.

**Robustness to Noise**    To test the robustness to noise in the observed data, we add Gaussian noise with different standard deviations ($\sigma = \{0.01, 0.05, 0.1\}$) to each training data point the model has access to and report the overall performance. As shown in Figure 3, while the performance of all methods drops as the noise level increases, SR-SCIENTIST consistently performs better than other methods, especially with Qwen3-Coder-480B and GLM-4.5-Air as LLM backbones.

**Symbolic Accuracy**    Table 2 shows the symbolic accuracy of representative methods, evaluated on all problems in the dataset. Overall, it is challenging to identify equations identical to the ground truth from observed data. Among these methods, SR-SCIENTIST achieves the best performance, correctly identifying the most equations. For a more qualitative assessment, Figure 4 presents the discovered equations for the PO10 and PO37 physics problems, which are related to nonlinear oscillators. For both problems, SR-SCIENTIST identifies the structure of the equation and its constants, while the equations from other methods are usually more complex and less accurate. Additionally, SR-SCIENTIST produces trajectories that illustrate its derivation process, providing information that can inspire human scientists to design better equations.

## 4.3 ANALYSIS

**Ablation studies**    Table 3 presents the results of our ablation studies on the data analysis and experience buffer components. The findings indicate that allowing the agent to perform data analysis is crucial for performance, particularly for the GPT-OSS-120B and Qwen3-Coder-480B models. Moreover, the experience buffer plays an important role in the long-term memory of the agent,

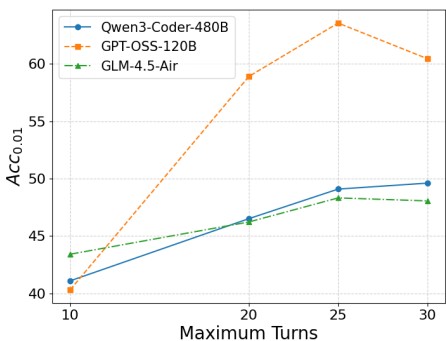

Figure 5: Overall performance under different maximum turns. We keep the total number of LLM calls at around 1,000 and trade off between maximum turns and iterations.

| Method | $Acc_{0.01}$ | $Acc_{0.001}$ |
|---|---|---|
| SR-SCIENTIST (GPT) | 63.57 | 49.35 |
| w/o $T_1$ | 35.66 | 16.28 |
| w/o experience | 57.36 | 41.86 |
| w/o top-k | 58.14 | 41.86 |
| SR-SCIENTIST (Qwen) | 49.09 | 24.55 |
| w/o $T_1$ | 41.08 | 14.73 |
| w/o experience | 35.66 | 16.28 |
| w/o top-k | 26.36 | 13.70 |
| SR-SCIENTIST (GLM) | 48.32 | 25.06 |
| w/o $T_1$ | 46.51 | 22.48 |
| w/o experience | 37.21 | 18.61 |
| w/o top-k | 41.96 | 21.71 |

Table 3: Ablation studies. In 'w/o $T_1$', the agent can not utilize the `data analyzer` tool. In 'w/o experience', the agent can not utilize the experience buffer and optimize from scratch for each iteration. In 'w/o top-k', we randomly sample equations from the experience buffer instead of using top-k sampling. GPT, Qwen, and GLM represent GPT-OSS-120B, Qwen3-Coder-480B, and GLM-4.5-Air, respectively.

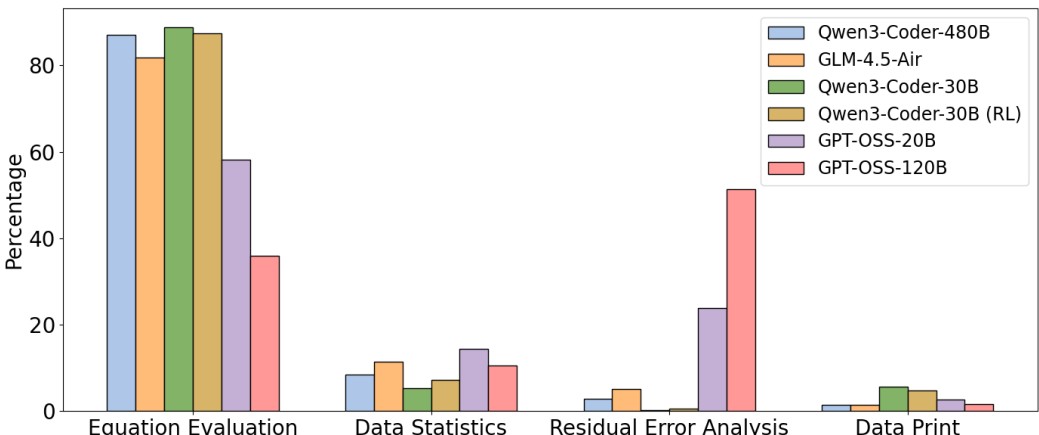

Figure 6: The tool call behaviors of different models.

allowing it to utilize the experience of top-performing equations from previous iterations. Either directly removing the experience buffer or randomly sampling equations from it leads to a significant performance drop.

**The Effect of the Turn Number** We fixed the LLM calls at around 1,000 and traded off between maximum turns and iterations to investigate the effect of different exploration lengths at each iteration. Increasing the maximum turns from 10 to 25, as shown in Figure 5, significantly improves performance, highlighting the value of long-horizon optimization. However, performance stagnates or slightly drops beyond 25 turns, indicating that excessively long exploration does not bring additional performance and that the inference budget should be allocated to initiating new iterations.

**Tool Call Analysis** We analyzed the distribution of tool call types and instructed LLMs to classify the `data analyzer` behaviors, presenting the results in Figure 6. Overall, the Qwen and GLM families exhibit similar patterns, with around 80% of their calls dedicated to equation evaluation and 20% to data analysis. Within the data analysis calls, data statistics generally account for a higher percentage, showing the importance of calculating relevant values such as correlations and averages.

In contrast, GPT-OSS-20B and GPT-OSS-120B tend to directly write their own code to perform residual error analysis for more fine-grained information. Further case studies show that GPT-OSS-120B also tends to directly define equation constants through data analysis, demonstrating greater flexibility. Moreover, after RL training, the Qwen3-Coder-30B also increases its use of data statistics for more advanced data analysis.

## 5 LIMITATIONS AND FUTURE DIRECTIONS

While SR-SCIENTIST demonstrates significant advantages in equation discovery, we identify several limitations that open avenues for future research. First, in our main experiments, we only used text-only models. Expanding the model to incorporate multi-modal inputs and include plots in the analysis phase is a promising direction. Second, in noisy scenarios, although SR-SCIENTIST performs best, it still suffers from a significant performance drop. Future work could explore incorporating common noise-handling strategies into the agent's prompt to guide its data analysis. Third, since we only provide the top-performing equations to the agent in each iteration, the agent may still re-explore equations that previously performed poorly. Optimizing the agent's memory system, such as by distilling high-level experience from the equations, is an interesting future direction.

## 6 CONCLUSION

In this paper, we introduce SR-SCIENTIST, a framework that transforms the LLM from a passive equation proposer into an autonomous scientist for symbolic regression. By analyzing data, evaluating and refining equations, the agent generates and refines hypotheses through active environmental interaction. Our experiments show that SR-SCIENTIST significantly outperforms existing methods in precision, generalization, robustness to noise, and symbolic accuracy. Furthermore, we develop a complete reinforcement learning pipeline that allows the agent to self-evolve and enhance its discovery capabilities.

## ACKNOWLEDGMENTS

We would like to express our great gratitude to Yan Ma for assisting with the writing and to Xuefeng Li for providing advice on RL Infra. This work was partially funded by the National Natural Science Foundation of China (62476168), the National High Technology Research and Development Program of China (2015AA015408), the Shanghai Science and Technology Development Funds (14ZR1403200), and the AI for Science Program of the Shanghai Municipal Commission of Economy and Informatization (2025-GZL-RGZN-BTBX-01013).

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

## APPENDIX

## A  BASELINE IMPLEMENTATION DETAILS

**GPLearn**   The open-source `gplearn`[5] library is a Python package built on scikit-learn that uses genetic programming to perform SR. We set the population size to 500, the number of generations to 200, and the tournament size to 20. The function set includes addition, subtraction, multiplication, division, sine, cosine, square root, logarithm, absolute value, negation, and inverse.

**End-to-End Symbolic Regression (E2E)**   E2E (Kamienny et al., 2022) is a method that uses a single deep learning model, often a Transformer, to directly predict a complete mathematical expression, including both its structure and numerical constants. We implement the method using the `symbolicregression` Facebook repository[6]. For this method, we set the maximum input points to 200 and the number of trees for refinement to 10.

**NeSymReS**   NeSymReS (Biggio et al., 2021) is a deep learning approach that uses a pre-trained Transformer model to directly predict a mathematical expression from data points. We use the `NeuralSymbolicRegressionThatScales`[7] repository to implement this method. We set the number of data points passed to the Transformer to 500 and the beam size (expression sampling size) to 32.

**Deep Symbolic Regression (DSR)**   DSR (Petersen et al., 2021) is a deep learning approach that uses a recurrent neural network and reinforcement learning to search for the mathematical expression that best fits a given dataset. We implement the method using the `deep-symbolic-optimization`[8] repository. For this method, we set the number of samples to 100,000, the batch size to 512, and the learning rate to 0.0005. The function set includes addition, subtraction, multiplication, division, sine, cosine, exponentiation, and logarithm, with expression lengths ranging from 1 to 9.

**Unified Deep Symbolic Regression (uDSR)**   uDSR (Landajuela et al., 2022) is a deep learning framework that unifies multiple symbolic regression strategies into a single modular system. It extends DSR by combining approaches like neural-guided search, pre-training, and linear models. We implement it using the `deep-symbolic-optimization` repository. For the genetic programming configuration, we set the population size to 100, generations to 20, crossover probability to 0.5, mutation probability to 0.5, tournament size to 5, and maximum mutation tree depth to 3. Other parameters are the same as DSR's. The function set is: add, sub, mul, div, sin, cos, exp, log, and poly.

---

[5]https://gplearn.readthedocs.io/en/stable/
[6]https://github.com/facebookresearch/symbolicregression
[7]https://github.com/SymposiumOrganization/NeuralSymbolicRegressionThatScales
[8]https://github.com/dso-org/deep-symbolic-optimization

**Python Symbolic Regression (PySR)**   The PySR Python package[9] (Cranmer, 2023) uses a powerful Julia backend to find simple, interpretable mathematical expressions that best fit the observed data. We set the number of iterations to 125, cycles per iteration to 550, populations to 15 with a population size of 33, maximum size to 30, and the randomization weight to 0.1. Its binary operators are $+, -, *, /,$ pow, and its unary operators[10] are $\exp, \log, $sqrt$, \sin, \cos$.

**Library-Augmented Symbolic Regression (LaSR)**   LaSR (Grayeli et al., 2024) uses LLMs to discover and evolve a library of abstract, natural-language concepts. We implement the method using the `LibraryAugmentedSymbolicRegression.jl`[11] repository. For this method, we set the number of iterations to 25, cycles per iteration to 550, populations to 10 with a population size of 33, maximum size to 30, and the maximum number of concepts to 20. The LLM operation weights for crossover, mutation, and randomization are each set to 0.02. The binary operators, unary operators, nested constraints, and other constraints are the same as those in PySR.

**LLM-SR**   LLM-SR (Shojaee et al., 2025a) leverages the extensive scientific knowledge and code-generation capabilities of LLMs to discover mathematical equations. It treats equations as programs and combines an LLM's scientific priors with an evolutionary search to find accurate and interpretable formulas. We implement it using the `llm-srbench`[12] repository with its default configuration and set the maximum number of LLM calls to 1,000 for each problem.

## B   DETAILS OF SR-SCIENTIST INFERENCE FRAMEWORK

For each iteration, we set the initial MAPE goal to 0.1% and the termination MAPE threshold to 0.0001%. For each LLM, we set the sampling temperature to 0.7 and the maximum completion length per call to 8,192. For the `equation evaluator` tool, to control the length and the complexity of the generated equations, we set the maximum number of parameters to 10 in all experiments, following Shojaee et al. (2025a). To ensure a fair comparison with methods like LLM-SR, the optimization goal for the BFGS algorithm is the mean square error. For parameter optimization, we use the `scipy` library for nonlinear optimization with the BFGS algorithm. The prompt for the LLM agent is shown in Figure 15 and Figure 16.

## C   DETAILS OF SR-SCIENTIST TRAINING FRAMEWORK

### C.1   TRAINING DATA SYNTHESIS

We synthesize our training data following the approach of Shojaee et al. (2025a), using a hybrid model-based and rule-based method. The problems cover four scientific disciplines: materials science (Stress with respect to Strain and Temperature), chemistry (Reaction rate with respect to Time and Concentration), biology (Growth rate with respect to Time and Population size), and physics (Acceleration with respect to Time, Displacement, and Velocity). The detailed procedure is as follows:

(1) *Equation Skeleton Generation*: Each equation skeleton contains at least one known and one novel term. The known terms refer to common concepts from an LLM's pre-training knowledge, and the novel terms refer to terms outside the LLM's prior knowledge. We synthesize both types of terms by prompting Claude 4 Sonnet. These terms are then combined into equation skeletons using addition. To ensure a moderate level of complexity, each equation is limited to 2-4 total terms. To prevent potential data leakage between our training set and the benchmark data, two authors of the paper independently identify equations that are identical or too similar and discuss them for further reconciliation. Any equations deemed too similar are discarded.

(2) *Parameter Instantiation*: For each equation skeleton, we transform it into a complete equation by assigning values to each constant. This process is not a random assignment; rather, it is

---

[9]https://astroautomata.com/PySR/

[10]Nested Constraints: $\sin : \{\sin : 0, \cos : 0\}, \cos : \{\sin : 0, \cos : 0\}, \exp : \{\exp : 0, \log : 0\}, \log : \{\exp : 0, \log : 0\}$, sqrt $: \{$sqrt$ : 0\}$; Constraints: $\{\sin : 10, \cos : 10, \exp : 20, \log : 20, $sqrt$ : 20, $pow$ : (-1, 20)\}$

[11]https://github.com/trishullab/LibraryAugmentedSymbolicRegression.jl

[12]https://github.com/deep-symbolic-mathematics/llm-srbench

| Method | Overall | | Material Science | | Chemistry | | Biology | | Physics | |
|---|---|---|---|---|---|---|---|---|---|---|
| | $Acc_{0.01}$ | $Acc_{0.001}$ | $Acc_{0.01}$ | $Acc_{0.001}$ | $Acc_{0.01}$ | $Acc_{0.001}$ | $Acc_{0.01}$ | $Acc_{0.001}$ | $Acc_{0.01}$ | $Acc_{0.001}$ |
| SR-SCIENTIST | 32.30 | 16.02 | 81.33 | 52.00 | 22.22 | 5.56 | 22.22 | 8.33 | 18.18 | 8.33 |
| + RL (log-linear) | 40.92 | 20.69 | 85.33 | 65.33 | 37.38 | 7.46 | 29.17 | 11.11 | 25.00 | 11.37 |
| + RL (stepwise) | 37.21 | 17.83 | 84.00 | 56.00 | 27.78 | 2.78 | 33.33 | 16.67 | 20.45 | 9.09 |

Table 4: The performance of different reward functions. 'log-linear' refers to the reward function corresponding to Equation 2 and 'stepwise' refers to the reward function corresponding to Equation 5.

guided by the scientific context and significance of each term. For example, in the context of material science, the reference temperature $T_0$ is set to 273.15 K (the triple point of water). We instruct the LLMs to perform value assignment, which is then followed by human validation.

(3) *Data Point Generation*: For the material science equations, which represent static systems, we generate data points by uniformly sampling across a defined range of parameters. The temperature $T$ is sampled from the range of $[273\ \text{K}, 573\ \text{K}]$, and strain $\epsilon$ is sampled from $[0, 0.6]$. A total of 5000 points are generated using an evenly spaced grid. The OOD test set is created by taking the 500 data points with the highest temperature values. The remaining 4500 points are then randomly shuffled, with 500 points allocated to the test set and the rest forming the training set. For the chemistry, biology, and physics equations, which represent dynamic systems, we use the `solve_ivp` function from the SciPy Python package with the RK45 method to solve the differential equations. Given initial conditions at $t = 0$, we solve the equations to obtain the relationship between variables over time (equations that cannot be solved are simply discarded). We generate 5000 data points by uniformly sampling the time variable $t$ from the range of $[0, 60]$. The OOD test set is created by selecting the 500 data points with the highest time values. The remaining 4500 points are randomly partitioned into a 500-point test set and the remaining training set.

(4) *Filtering and Final Dataset Assembly*: In the filtering stage, we compute the statistical properties of the resulting data. Equations with data points that are scientifically anomalous are filtered out. This process ensures that our final dataset contains only scientifically meaningful equations with plausible data points. Finally, the instantiated equations, their corresponding numerical data points, and other relevant information (variable symbols, equation names, etc.) are packed for the subsequent training.

## C.2 REWARD DESIGN

Besides the log-linear reward design, we also explore the stepwise reward function as follows:

$$\mathcal{R} = \begin{cases} 1.0 & \text{if } s < 0.001 \\ 0.5 & \text{if } 0.001 \le s < 0.01 \\ 0.25 & \text{if } 0.01 \le s < 0.1 \\ 0.1 & \text{if } 0.1 \le s < 1 \\ 0.0 & \text{if } s \ge 1 \end{cases} \tag{5}$$

where $s$ denotes the MAPE value. As shown in Table 4, the log-linear reward function performs better than the stepwise function.

## C.3 TRAINING DETAILS

We conduct RL training on 32 NVIDIA H200 GPUs. For the infrastructure, we use the `verl`[13] framework, employing SGLang as the rollout engine and FSDP as the training engine. For rollouts, we implement batch-level asynchronous rollouts to reduce the rollout time. We set the temperature to 1.0, the maximum response length to 10,240, and the maximum number of turns to 20. We use

---

[13] https://github.com/volcengine/verl

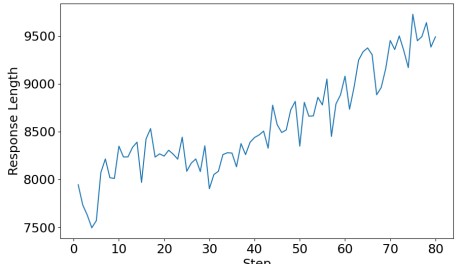

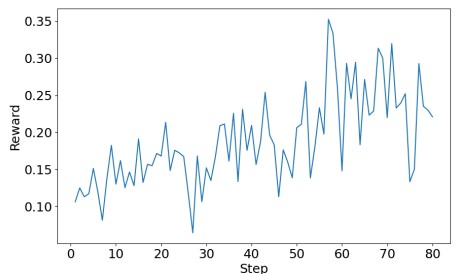

Figure 7: The change in response length during training.

Figure 8: The change in reward score during training.

| Method | Overall | | Material Science | | Chemistry | | Biology | | Physics | |
|---|---|---|---|---|---|---|---|---|---|---|
| | $Acc_{0.01}$ | $Acc_{0.001}$ | $Acc_{0.01}$ | $Acc_{0.001}$ | $Acc_{0.01}$ | $Acc_{0.001}$ | $Acc_{0.01}$ | $Acc_{0.001}$ | $Acc_{0.01}$ | $Acc_{0.001}$ |
| *Qwen3-32B* | | | | | | | | | | |
| LLM-SR | 30.49 | 9.04 | 34.67 | 18.67 | 38.89 | 5.56 | 15.28 | 4.17 | 29.55 | 9.09 |
| SR-SCIENTIST | 29.46 | 15.50 | 72.00 | 48.00 | 22.22 | 8.33 | 20.83 | 4.17 | 15.91 | 9.09 |
| +RL | 39.53 | 20.16 | 80.00 | 64.00 | 38.89 | 8.33 | 25.00 | 4.17 | 25.00 | 13.64 |
| *Qwen3-14B* | | | | | | | | | | |
| LLM-SR | 27.91 | 9.30 | 56.00 | 32.00 | 16.67 | 2.78 | 8.33 | 0.00 | 31.82 | 6.82 |
| SR-SCIENTIST | 18.60 | 6.98 | 48.00 | 28.00 | 13.89 | 0.00 | 12.50 | 4.17 | 9.09 | 2.27 |
| + RL | 23.26 | 10.08 | 52.00 | 40.00 | 16.67 | 2.78 | 20.83 | 0.00 | 13.64 | 4.55 |
| *Qwen3-8B* | | | | | | | | | | |
| LLM-SR | 22.48 | 8.53 | 48.00 | 28.00 | 16.67 | 5.56 | 8.33 | 0.00 | 20.45 | 4.55 |
| SR-SCIENTIST | 15.50 | 4.65 | 40.00 | 16.00 | 8.33 | 5.56 | 16.67 | 0.00 | 6.82 | 0.00 |
| + RL | 20.93 | 9.30 | 68.00 | 44.00 | 16.67 | 0.00 | 4.17 | 0.00 | 6.82 | 2.27 |

Table 5: Performance comparison of different LLM backbones for RL training, including Qwen3-8B, Qwen3-14B, and Qwen3-32B.

a prompt batch size of 32 and generate 8 rollouts per prompt. The KL loss coefficient is set to 0 to encourage exploration. The remaining parameters use the default settings.

Figures 7 and 8 present the training dynamics. As shown, the reward progressively increases and saturates at around 60 steps. The response length also increases, demonstrating that the model learns a long-horizon problem-solving strategy.

### C.4 THE EFFECT OF LLM BACKBONE

To further investigate the effect of RL training on different models, we conduct RL training on the Qwen3-8B, Qwen3-14B, and Qwen3-32B models using the same training configuration detailed in Appendix C.3 and present the results in Table 5. All models achieve significant performance gains after RL training, with most improving their performance across all subjects. When comparing LLM-SR and SR-SCIENTIST without additional RL training, SR-SCIENTIST outperforms LLM-SR on $Acc_{0.01}$ and matches it on $Acc_{0.1}$ when using Qwen3-32B as the LLM backbone. However, it lags behind when using the Qwen3-8B and Qwen3-14B models. This highlights the importance of a model's intrinsic ability in a framework that provides the model with greater autonomy.

## D SUPPLEMENTARY ANALYSIS AND EVALUATION DETAILS

### D.1 ANALYSIS FOR THE EVALUATION METRICS

Table 7 presents the results measured by the $R^2$ metric. As shown, the differences between the various methods are small, making it difficult to distinguish their performance. Additionally, we present the NMSE results in Table 6. The average NMSE is sensitive to maximum values, which makes it

| Method | NMSE (Average) | NMSE (Max) | NMSE (Min) | NMSE (Medium) |
|---|---|---|---|---|
| LaSR | 1.53e-02 | 1.67e-01 | 2.66e-13 | 1.84e-03 |
| LLM-SR | 3.19e-03 | 7.83e-02 | 1.72e-14 | 8.87e-06 |
| SR-SCIENTIST | 1.29e-03 | 4.06e-02 | 1.57e-14 | 2.64e-06 |

Table 6: Comparison of SR-SCIENTIST and LLM-based SR baseline methods measured by NMSE. We report the result of Qwen3-Coder-480B on the physics subject.

| Method | Overall ($R^2$) | Material Science ($R^2$) | Chemistry ($R^2$) | Biology ($R^2$) | Physics ($R^2$) |
|---|---|---|---|---|---|
| LaSR | 0.991 | 0.998 | 0.996 | 0.988 | 0.985 |
| LLM-SR | 0.998 | 0.996 | 1.000 | 1.000 | 0.997 |
| SR-SCIENTIST | 0.998 | 0.996 | 1.000 | 0.995 | 0.999 |

Table 7: Comparison of SR-SCIENTIST and LLM-based SR baseline methods measured by $R^2$. We report the result of Qwen3-Coder-480B.

an unreliable indicator of overall performance. Although quantitative methods could be employed, such as truncating errors that exceed a certain constant, this threshold is difficult to define. Using the median value is also insufficient, as it cannot account for the entire performance distribution. Therefore, we abandoned these metrics and instead used accuracy-to-tolerance as the main metric.

## D.2 COST

For LLM-based methods, the cost mainly comes from API calls to LLMs. In Table 8, we present the estimated cost of SR-SCIENTIST with GPT-OSS-120B and GPT-OSS-20B as the LLM backbones. We calculated the cost for each problem based on token consumption and common API pricing. As shown, the cost is acceptable for practical usage, and when considering cached tokens, the price can be further reduced. Additionally, we deployed GPT-OSS-120B on a local server with 2 NVIDIA H100s for batch-level inference on 129 problems and recorded the wall-clock times. The maximum time was no more than 5 hours, a duration that is acceptable for practical usage, as typical scenarios involve only a few problems.

## D.3 OOD PERFORMANCE OF OTHER MODELS

We illustrate the OOD performance using other models as LLM backbones in Figures 9, 10, 11, and 12. For models including GLM-4.5-Air, GPT-OSS-120B, and GPT-OSS-20B, SR-SCIENTIST consistently outperforms the other methods in overall accuracy and most subjects. For Qwen3-Coder-30B, it slightly lags behind the method PySR on the OOD data. After RL training, it not only enhances its performance on ID data but also on OOD data, showing the generalization of the discovered equations.

## D.4 OTHER QUALITATIVE RESULTS

To demonstrate how the agent gains insight from data analysis and long-horizon exploration, we provide two snippets of the agent trajectories in Figure 13 and Figure 14. As shown by the reasoning process in the dashed box, the agent forms its hypothesis for the equation based on its analysis of the data through code, along with the performance of historical equations. This analysis grounded in the data helps decrease the search space compared to previous methods. The reasoning and analysis process can also inspire human scientists towards better equation design.

## D.5 SENSITIVITY ANALYSIS OF HYPERPARAMETERS

We present a sensitivity analysis of the hyperparameters in Table 9 and Table 10, covering the number of equations fetched from the experience buffer and the initial MAPE goal. Regarding the number of fetched equations, our method remains robust when this value is relatively small. However,

| Method | API Price | API Cost Per Problem |
|--------|-----------|----------------------|
| SR-SCIENTIST (GPT-OSS-120B) | Input: $0.05/1M; Output: $0.25/1M | $0.25 |
| SR-SCIENTIST (GPT-OSS-20B) | Input: $0.03/1M; Output: $0.15/1M | $0.1 |

Table 8: The estimated cost of SR-SCIENTIST. This calculation does not consider cached tokens; including them would reduce the cost further.

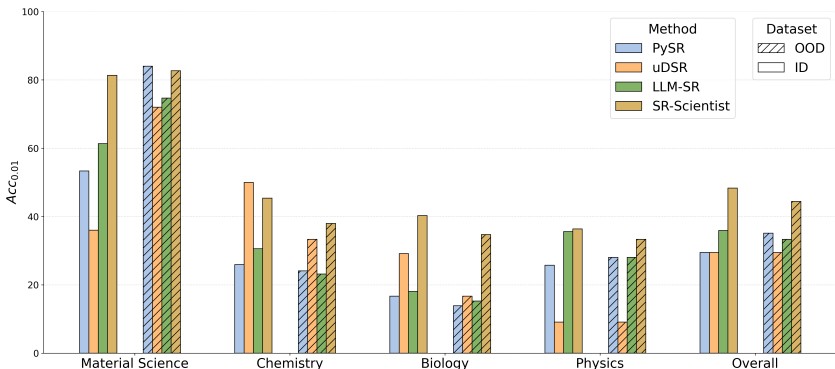

Figure 9: Detailed results of in-domain (ID) and out-of-domain (OOD) performance using $\text{Acc}_{0.01}$ across various LSR-Synth scientific domains. (with GLM-4.5-Air as LLM backbone)

| $K$ | $\text{Acc}_{0.01}$ | $\text{Acc}_{0.001}$ |
|-----|---------------------|----------------------|
| 1 | 63.57 | 49.61 |
| 3 | 63.57 | 49.35 |
| 5 | 63.31 | 50.90 |
| 10 | 61.50 | 47.29 |

| $s_{\text{goal}}$ | $\text{Acc}_{0.01}$ | $\text{Acc}_{0.001}$ |
|-------------------|---------------------|----------------------|
| 0.1 | 64.34 | 48.84 |
| 0.01 | 63.31 | 49.61 |
| 0.001 | 63.57 | 49.35 |
| 0.0001 | 51.74 | 37.70 |

Table 9: Sensitivity analysis of the number of equations $K$ fetched from the experience buffer. (with GPT-OSS-120B as LLM backbone)

Table 10: Sensitivity analysis of the initial MAPE goal $s_{\text{goal}}$. (with GPT-OSS-120B as LLM backbone)

performance declines slightly when it is increased to 10. This suggests that including a large number of fetched equations in the context window may strain the model's capability. Regarding the initial MAPE goal, performance is insensitive to relatively easy targets, as our dynamic goal update mechanism effectively prevents trivial goals. However, an extremely challenging goal (e.g., 0.0001) leads to degraded performance, causing the model to become trapped during the optimization process.

### D.6 SYMBOLIC ACCURACY CALCULATION

We calculate symbolic accuracy using a two-step evaluation strategy: an initial assessment by an LLM, followed by human verification. Specifically, we instruct GPT-OSS-120B to determine the equivalence between a predicted equation and its ground truth counterpart using the prompt shown in Figure 17. To ensure reliability, the LLM evaluates each problem 10 times. Our preliminary studies on 121 cases reveal a 98.3% consistency rate between LLM and human evaluations for instances where the 10 voting results were unanimous. Therefore, our practical workflow involves first using the LLM for evaluation. However, for any case where the LLM's votes are inconsistent, we pass the problem to human evaluators for a final decision.

### D.7 TOOL CALL ANALYSIS

We instruct GPT-OSS-120B to perform tool call analysis using the prompt in Figure 18. We check 54 cases and find a 94.4 % consistency between human and LLM-based evaluation.

# E  THE USE OF LARGE LANGUAGE MODELS

The large language model was utilized as a writing assistant during the preparation of this manuscript. Its role was strictly limited to proofreading for grammatical errors, improving sentence structure, and enhancing readability. The large language model was not used for generating any core ideas, methodology, or scientific content. The authors take full responsibility for all content presented.

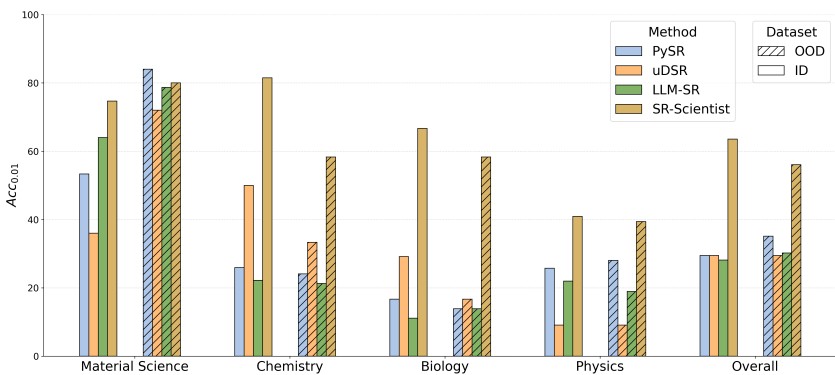

Figure 10: Detailed results of in-domain (ID) and out-of-domain (OOD) performance using $Acc_{0.01}$ across various LSR-Synth scientific domains. (with GPT-OSS-120B as LLM backbone)

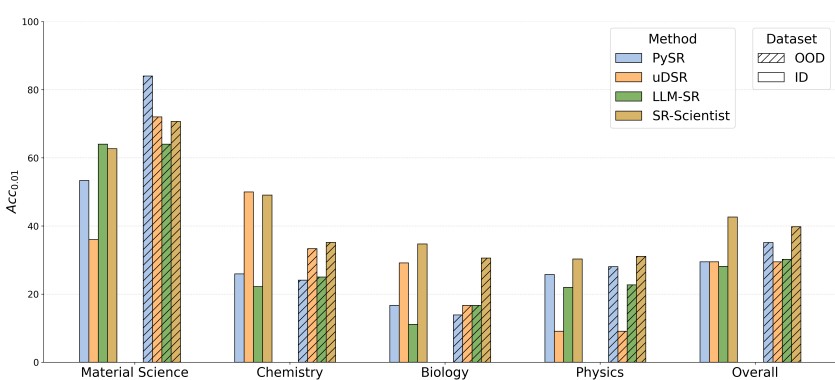

Figure 11: Detailed results of in-domain (ID) and out-of-domain (OOD) performance using $Acc_{0.01}$ across various LSR-Synth scientific domains. (with GPT-OSS-20B as LLM backbone)

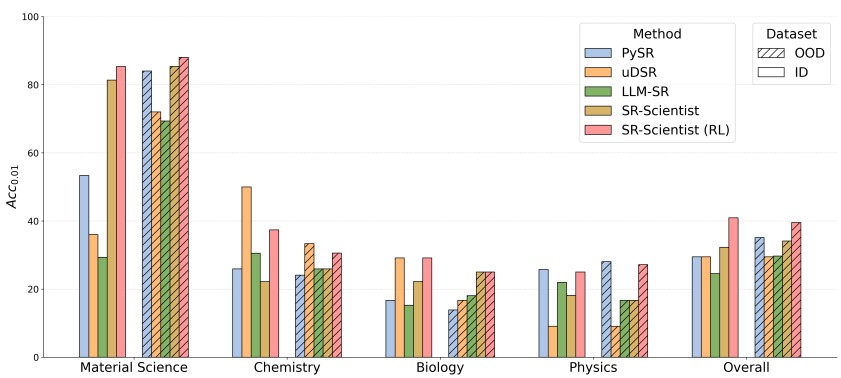

Figure 12: Detailed results of in-domain (ID) and out-of-domain (OOD) performance using $Acc_{0.01}$ across various LSR-Synth scientific domains. (with Qwen3-Coder-30B as LLM backbone)

```
 Agent: Now I'll analyze the data further to understand the relationship better. Let me look at how the
         growth rate changes with respect to population and time.
         ```
         import json
         import sys
         import numpy as np

         # load the data
         input_data_str = sys.stdin.read()
         data_list = json.loads(input_data_str)

         # Convert to numpy arrays
         data_array = np.array(data_list)
         outputs = data_array[:, 0]  # Population growth rate
         t_values = data_array[:, 1]  # Time
         P_values = data_array[:, 2]  # Population

         # Look at the pattern when population is high (near carrying capacity)
         high_pop_mask = P_values > 90
         high_pop_data = data_array[high_pop_mask]
         print(f"Number of high population (>90) data points: {len(high_pop_data)}")
         if len(high_pop_data) > 0:
             high_pop_growth_rates = high_pop_data[:, 0]
             print(f"High population growth rates: min={np.min(high_pop_growth_rates):.6f},
             max={np.max(high_pop_growth_rates):.6f}, mean={np.mean(high_pop_growth_rates):.6f}")

         # Look at the pattern when population is low
         low_pop_mask = P_values < 20
         low_pop_data = data_array[low_pop_mask]
         print(f"Number of low population (<20) data points: {len(low_pop_data)}")
         if len(low_pop_data) > 0:
             low_pop_growth_rates = low_pop_data[:, 0]
             print(f"Low population growth rates: min={np.min(low_pop_growth_rates):.6f},
             max={np.max(low_pop_growth_rates):.6f}, mean={np.mean(low_pop_growth_rates):.6f}")

         # Look at early time vs late time patterns
         early_time_mask = t_values < 10
         late_time_mask = t_values > 40
         early_data = data_array[early_time_mask]
         late_data = data_array[late_time_mask]

         print(f"Early time growth rates: min={np.min(early_data[:, 0]):.6f},
         max={np.max(early_data[:, 0]):.6f}, mean={np.mean(early_data[:, 0]):.6f}")
         print(f"Late time growth rates: min={np.min(late_data[:, 0]):.6f},
         max={np.max(late_data[:, 0]):.6f}, mean={np.mean(late_data[:, 0]):.6f}")

         # Check if there's a quadratic relationship
         P_squared = P_values ** 2
         correlation_p2_o = np.corrcoef(P_squared, outputs)[0, 1]
         print(f"Correlation between P^2 and Growth Rate: {correlation_p2_o:.6f}")
         ```

 Tool:   Number of high population (>90) data points: 3468
         High population growth rates: min=-0.051324, max=5.084093, mean=0.118920
         Number of low population (<20) data points: 234
         Low population growth rates: min=1.016704, max=16.429510, mean=6.147035
         Early time growth rates: min=0.792945, max=24.778742, mean=10.340174
         Late time growth rates: min=-0.048202, max=0.077139, mean=0.008463
         Correlation between P^2 and Growth Rate: -0.704877
-----------------------------------------------------------------------------------------------------
 Agent: Based on my analysis, I can see that this appears to be a population growth model with
         characteristics of logistic growth: High population values (>90) have very low growth rates,
         approaching zero. Low population values (<20) have high growth rates. Early times have higher
         growth rates than late times. There's a strong negative correlation between P^2 and growth rate.
         This suggests a logistic growth model where the growth rate decreases as population approaches a
         carrying capacity. Let me try a more refined logistic model:
-----------------------------------------------------------------------------------------------------
         def equation(t: np.ndarray, P: np.ndarray, params: np.ndarray) -> np.ndarray:
             r, K, alpha, beta = params[0], params[1], params[2], params[3]

             # Ensure parameters stay in reasonable ranges
             K = np.abs(K) + 90.0  # Near maximum observed population
             alpha = np.abs(alpha)  # Ensure positive decay rate

             logistic_term = r * P * (1 - P / K)
             time_decay = np.exp(-alpha * t)
             output = logistic_term * time_decay + beta

             return output
```

Figure 13: The snippet of the agent trajectory for the BPG18 biology problem. (with Qwen3-Coder-480B as LLM backbone)

```
Agent: We need to improve model. Let's inspect data to see relationships. Use data_analyzer to view
        stats.

        ```
        import json, sys, numpy as np
        data_str = sys.stdin.read()
        data = json.loads(data_str)
        # Convert to arrays
        outputs = np.array([row[0] for row in data])
        inputs = np.array([row[1:] for row in data])
        X = inputs.T
        x, t, v = X
        print('samples', len(outputs))
        print('output stats: mean', outputs.mean(), 'std', outputs.std())
        print('x stats: mean', x.mean(), 'std', x.std())
        print('t stats: mean', t.mean(), 'std', t.std())
        print('v stats: mean', v.mean(), 'std', v.std())
        # compute correlation
        corr = np.corrcoef(np.vstack([outputs, x, t, v]))
        print('Correlation matrix:\n', corr)
        # maybe try simple linear regression coefficients via least squares to see which terms matter
        A = np.column_stack([x, t, v, np.ones_like(x)])
        coef, *_ = np.linalg.lstsq(A, outputs, rcond=None)
        print('Linear coeffs (x,t,v,const):', coef)
        ```

Tool:   samples 4000
        output stats: mean -0.13570954609178262 std 7.8811164966115
        x stats: mean -0.04767970374011202 std 0.7673236558313943
        t stats: mean 18.040196053244408 std 10.417019343399922
        v stats: mean -0.024004985556995963 std 2.4674150781331674
        Correlation matrix:
        [[ 1. -0.99249334 -0.01201174 0.00565316][-0.99249334 1. 0.01074399 -0.00662076]
        [-0.01201174 0.01074399 1. 0.01175548] [ 0.00565316 -0.00662076 0.01175548 1. ]]
        Linear coeffs (x,t,v,const): [-1.01937290e+01 -1.01219659e-03 -2.88143544e-03 -6.03552468e-01]
--------------------------------------------------------------------------------------------------
Agent: The data analysis shows high correlation: output(acceleration) strongly negatively correlated
        with x (position) -0.9925 correlation. t correlation also -0.9925? Actually correlation matrix
        shows output vs t correlation -0.9925 (since second row/column is t). Wait matrix: rows order:
        outputs, x, t, v? They stacked [outputs, x, t, v]. So correlation[0,2] is output vs t = -0.9925.
        Also output vs x = -0.9925? Actually correlation[0,1] is -0.9925 as well. So both x and t highly
        correlated with output, likely because time and position are linearly related (maybe x
        ˜ something of t). The simple linear regression gave coeffs: x: -10.1937, t: -0.001012,
        v: -0.00288, const: -0.6035. The magnitude of x coefficient is large. Maybe output is
        approximated by -10 * x + small other terms. The correlation is near -1, suggests a linear
        relationship with x (and t maybe similar). So maybe acceleration = -k * x (like simple harmonic
        oscillator). For a non-linear oscillator, could have term -kx - alphax^3 but correlation near -1
        suggests linear dominates. Our previous simple cubic model gave huge MAPE, because we didn't
        include intercept and maybe scaling. Let's try a linear model: a = p0 * x + p1 (constant).
        Use params[0] * x + params[1]. That may achieve low error.
        Let's test.
--------------------------------------------------------------------------------------------------
        import numpy as np
        def equation(x: np.ndarray, t: np.ndarray, v: np.ndarray, params: np.ndarray) -> np.ndarray:
            """Simple linear model: a = p0 * x + p1"""
            return params[0] * x + params[1]
```

Figure 14: The snippet of the agent trajectory for the PO29 physics problem. (with GPT-OSS-120B as LLM backbone)

```python
Your goal is to determine the correct equation, implement it as a Python function, and optimize it until
the mean absolute percentage error is less than 0.1%.
You should use the `equation_evaluator` tool to evaluate the equation's goodness of fit and the
`data_analyzer` tool to write code for data analysis.
For the `equation_evaluator`, it is a code interpreter that wraps your function with the following code:
```python
import numpy as np
import sys
import json
# Initialize parameters
MAX_NPARAMS = 10
params = [1.0] * MAX_NPARAMS
# Example of a user-provided equation
def equation(x: np.ndarray, t: np.ndarray, v: np.ndarray, params: np.ndarray) -> np.ndarray:
    """
    Mathematical function for Acceleration in Nonl-linear Harmonic Oscillator
    Args:
    x: A numpy array representing observations of Position at time t.
    t: A numpy array representing observations of Time.
    v: A numpy array representing observations of Velocity at time t.
    params: Array of numeric constants or parameters to be optimized
    Return:
        A numpy array representing Acceleration in Nonl-linear Harmonic Oscillator as the result of
        applying the mathematical function to the inputs.
    """
    output = params[0] * x + params[1] * t + params[2] * v + params[3]
    return output
def evaluate(data: list) -> float:
    # Load data observations
    outputs = np.array([row[0] for row in data])
    inputs = np.array([row[1:] for row in data])
    X = inputs
    # Optimize parameters based on data
    from scipy.optimize import minimize
    def loss(params):
        y_pred = equation(*X.T, params)
        return np.mean((y_pred - outputs) ** 2)
    loss_partial = lambda p: loss(p)
    result = minimize(loss_partial, [1.0] * MAX_NPARAMS, method='BFGS')
    # Return evaluation score
    optimized_params = result.x
    mse = result.fun
    if np.isnan(mse) or np.isinf(mse):
        return None, None, None
    var_outputs = np.var(outputs)
    if np.isclose(var_outputs, 0):
        nmse = 0.0 if np.isclose(mse, 0) else np.inf
    else:
        nmse = mse / var_outputs
    y_pred = equation(*X.T, optimized_params)
    zero_mask = np.isclose(outputs, 0)
    non_zero_mask = ~zero_mask
    mape = 0.0
    if np.any(non_zero_mask):
        relative_errors=np.abs((y_pred[non_zero_mask] - outputs[non_zero_mask])/outputs[non_zero_mask])
        mape = np.mean(relative_errors)
    return float(mse), float(nmse), float(mape)
if __name__ == '__main__':
    input_data_str = sys.stdin.read()
    data_list = json.loads(input_data_str)
    mse, nmse, mape = evaluate(data_list)
    if mse is not None:
        print(f"MSE:{{mse:.6e}};NMSE:{{nmse:.6e}};Mean absolute percentage error:{{mape:.4\%}}")
        if mape < {mape_threshold}:
            print("Success: The mean absolute percentage error is smaller than 0.1%.")
        else:
            print("Failure: The mean absolute percentage error is larger than 0.1%.")
```
As shown, the `equation_evaluator` tool assesses your equation's goodness of fit. It uses SciPy's BFGS
optimizer to find the optimal parameters for your equation based on the dataset. It then provides an
output with performance metrics (Mean Squared Error, Normalized Mean Squared Error, and Mean
Absolute Percentage Error), the success status, and details of any bugs.
In utilizing the tool, you only need to pass the entire function, including the function header, to the
tool.
For the `data_analyzer` tool, it is a code interpreter that can run your data exploration snippet.
You can access the data as shown in the example.
However, you are forbidden from using any libraries like Matplotlib for plotting figures for analysis.
```python
import json
import sys
# load the data
input_data_str = sys.stdin.read()
data_list = json.loads(input_data_str)
# print the first 5 entries
# In each entry of data_list, the first value is the output to predict, and the rest are the inputs.
print(data_list[:5])
"""
```

Figure 15: System prompt for the agent.

```
Find the mathematical function skeleton that represents Acceleration in Nonl-linear Harmonic
Oscillator, given data on Position at time t, Time, and Velocity at time t.

Follow these steps to solve the problem:

**1. Implement the Equation in Code**

* Based on your knowledge and analysis, identify the standard equation and implement it in the code.
* Your equation will likely have one or more constants. Use elements from the `params` list
(e.g., `params[0]`, `params[1]`, `params[2]`) to represent these constants, as the `equation_evaluator`
tool is designed to optimize them. Note that the `params` list has a fixed size of 10
(`MAX_NPARAMS = 10`), so you can use up to 10 parameters in your model.

**2. Test, Analyze, and Refine**
    * Evaluate the equation's goodness of fit using the `equation_evaluator` tool.
    You need to pass the entire function, including the function header, to the tool. Here is an example:
```python
def equation(x: np.ndarray, t: np.ndarray, v: np.ndarray, params: np.ndarray) -> np.ndarray:
""" Mathematical function for Acceleration in Nonl-linear Harmonic Oscillator
    Args:
        x: A numpy array representing observations of Position at time t.
        t: A numpy array representing observations of Time.
        v: A numpy array representing observations of Velocity at time t.
        params: Array of numeric constants or parameters to be optimized
    Return:
        A numpy array representing Acceleration in Nonl-linear Harmonic Oscillator as the result
        of applying the mathematical function to the inputs.
"""
    output = params[0] * x + params[1] * t + params[2] * v + params[3]
    return output
```
    You can modify the function body, but the function header must remain unchanged.
    * Your goal is to reduce the mean absolute percentage error to less than 0.1000%.
    Meeting this condition indicates that your equation is a good fit for the data.
    * If this goal is not met, refine your equation in Python and observe its performance.
    You can write your own data exploration snippet and use the `data_analyzer` tool to execute it,
    allowing you to inspect the data for potential relationships or anomalies.
**3. Submit Your Final Answer**
    * Once you are confident your equation has met the condition, or if you conclude after numerous
    attempts that you cannot meet it, provide the completed Python function as your answer.
"""
```

Figure 16: User prompt for the agent.

```
Given the ground truth mathematical expression A and the hypothesis B, determine if there exist any
constant parameter values that would make the hypothesis equivalent to the given ground truth expression.
Let's think step by step. Explain your reasoning and then provide the final answer as:
{{ "reasoning": "Step-by-step analysis", "answer": "Yes/No" }}

Ground Truth A: {gt_equation}
Hypothesis B: {pred_equation}
```

Figure 17: Prompt for symbolic assessment.

```
You are an expert code analyst. Your task is to classify Python code snippets written by an AI agent that
is trying to discover scientific equations from data.
Analyze the primary purpose of the code snippet and assign it a category based on the definitions below.
Category Definitions:
- Data Statistics: The code calculates a statistical property of the data. Examples include
calculating the mean, variance, correlation, minimum, or maximum values.
  - IMPORTANT: For this category, you MUST respond in the format: Data Statistics: [STATISTIC_NAME].
    For example: Data Statistics: Correlation or Data Statistics: Mean.
- Data Print: The code prints raw data samples for the purpose of initial inspection and understanding.
A common example is printing the first few rows.
- Residual Error Analysis: The code evaluates a proposed mathematical equation by calculating the
residuals (the difference between predicted and actual values) or other error metrics to determine its
goodness of fit to the data.
If the code's purpose does not fit any of the categories listed above, create a new, concise, and
specific category name.
For all categories except 'Data Statistics', respond with ONLY the category name.
Classify the following Python code snippet.
Code Snippet:
---
{code}
---
```

Figure 18: Prompt for tool call analysis.

