# OpenReview forum: "SR-Scientist: Scientific Equation Discovery With Agentic AI"
_ICLR.cc/2026/Conference — ICLR 2026 Poster_

### Official Review · Reviewer_ewF1 · 2025-11-01

**Soundness:** 2
**Presentation:** 3
**Contribution:** 2
**Rating:** 4
**Confidence:** 3

**Summary:**

The paper introduces SR-Scientist, an agentic framework for scientific equation discovery through data-analysis and equation-evaluation tool calling. The authors also develop an RL training pipeline that improves agent performance. The method outperforms baselines on LLM-SRBench [1].

[1] Shojaee et al. "LLM-SRBench: A new benchmark for scientific equation discovery with large language models." (2025).

**Strengths:**

- The method is simple and clear. The task of scientific equation discovery is important, and the problem formulation is concise.
- SR-Scientist consistently outperforms state-of-the-art baselines on all models and across all tested scientific fields. The selected baselines make sense.
- The paper successfully demonstrates a path forward towards self improvement of the agent through an end-to-end RL pipeline.

**Weaknesses:**

- The work appears to be heavily inspired by SR-LLM [2]. Several key components, such as the equation-evaluator tool or the experience buffer, closely resemble those introduced in [2]. While the reframing of the problem as an agentic framework is probably useful, it is not a novel idea and has been widely used in many recent works.
- Results are evaluated solely on synthetic data (LSR-Synth, a synthetic subset of [1]). This limits the assessment of the method for practical applications. Adding experiments on real-world datasets or use cases where data often includes noise, outliers, etc., could provide a stronger evidence. Also, designing new tools to handle such challenges would strengthen the paper.
- The current design of the experience buffer ablation (Table 3) only confirms that accumulating knowledge over long horizons is better than constantly starting from scratch. Instead, this ablation should isolate the contribution of the suggested fetch-top-K buffer (for example, against a baseline that randomly selects equations).

[2] Shojaee et al. "LLM-SR: Scientific equation discovery via programming with large language models." (2024).

**Questions:**

1. Have the authors considered the use of broader tools for data analysis, such data visualization or graphical analysis? Do you think that expanding the framework's tool capabilities could be useful?
2. Could the authors please elaborate on the mechanisms that allow the framework to obtain a long-horizon behavior?
3. Do the authors plan to evaluate their method on real-world datasets?

---

> ### Author Response · Authors · 2025-11-20
> **Response to Reviewer ewF1 (1/2)**
>
> Thank you for your thoughtful review and constructive feedback. We appreciate your positive comments on our method's simplicity and strong performance, as well as the effectiveness of the RL component. We have provided a point-by-point reply to address your concerns below.
>
> > **Weakness 2:** Results are evaluated solely on synthetic data (LSR-Synth, a synthetic subset of [1]). This limits the assessment of the method for practical applications. Adding experiments on real-world datasets or use cases where data often includes noise, outliers, etc., could provide a stronger evidence. Also, designing new tools to handle such challenges would strengthen the paper.
>
> > **Question 3:** Do the authors plan to evaluate their method on real-world datasets?
>
>
> **Response:**  Thank you for your feedback and suggestions. We agree that evaluating our method on datasets that closely mirror real-world scenarios is crucial for assessing its practical applicability in equation discovery. In this regard, we would like to emphasize three key points:
>
> First, we have included noise robustness experiments to simulate real-world use cases. For the noise robustness experiments (Page 8, Lines 406-410), we added Gaussian noise with different standard deviations to the observed data. SR-Scientist consistently performs better than other methods, demonstrating its robustness to noise.
>
> Second, we wish to highlight that the LSR-Synth dataset is grounded in realistic scientific contexts through the following measures:
> - All equations, along with visualizations of their generated data points, were cross-checked by two subject matter experts to validate their scientific rigor.
> - All constants and variable ranges within the equations are designed to fit plausible physical scenarios.
> - Each equation is composed of both known and synthetic terms. Known terms refer to those commonly found in scientific equations and are well-documented in the literature for that domain and specific problem. These terms are included to ensure that the problems remain grounded in the established scientific context.
>
> Third, following the reviewer's suggestion, we have conducted new experiments on the Stress-Strain dataset [2-3]. This benchmark problem leverages a real-world experimental dataset, comprising tensile tests on Aluminum 6061-T651 across a range of temperatures. The inclusion of this benchmark helps test the performance on real-world experimental data. Moreover, there is no predetermined theoretical model structure for this problem, thereby avoiding memorization issues. The results are as follows:
>
> | Method | $\text{Acc}_{0.1}$| $\text{Acc}_{0.01}$|
> |---|---| ---|
> | PySR | 70.67 | 7.91 |
> | LaSR |  63.80 | 7.21 |
> | LLM-SR | 68.52| 8.74 |
> | SR-Scientist | 76.98 |16.02 |
>
> As shown, SR-Scientist significantly outperforms other methods, demonstrating its effectiveness on real-world data.  **We have added the clarification regarding the LSR-Synth dataset (Page 5, Lines 250-252) and included the new experiment (Page 8, Lines 429-431) in the revised manuscript.**
>
> > **Weakness 1:** The work appears to be heavily inspired by SR-LLM. Several key components, such as the equation-evaluator tool or the experience buffer, closely resemble those introduced in. While the reframing of the problem as an agentic framework is probably useful, it is not a novel idea and has been widely used in many recent works.
>
> **Response:** Thank you for this comment. We would like to point out that our method differs from previous work (e.g., LLM-SR) on symbolic regression (SR) or equation discovery in three key aspects:
>
> - Integration of Data Analysis Tools. Existing LLM-based SR methods lack direct analysis of observed data through tools to gain insights. By introducing data analysis tools, our method shifts the paradigm from enumerative search to data-driven reasoning and analysis. This is particularly critical for discovering novel equations where prior knowledge is limited.
> - Flexible Agentic Framework. Unlike previous frameworks that rely on fixed workflows, our agentic framework empowers the LLM to dynamically determine its next action based on environmental feedback. While agentic LLMs exist in fields like software engineering and search, their integration into equation discovery remains underexplored. Moreover, adapting general-purpose agentic models to the SR field is non-trivial, as it requires a specialized memory mechanism tailored for long-horizon continuous optimization.
> - Comprehensive Training Pipeline. Most existing LLM-based SR approaches focus solely on inference, overlooking the potential of task-specific training. We provide a comprehensive pipeline, ranging from training data construction to reward design, enabling the LLM to enhance its capabilities through RL.

---

> ### Author Response · Authors · 2025-11-20
> **Response to Reviewer ewF1 (2/2)**
>
> > **Weakness 3:** The current design of the experience buffer ablation (Table 3) only confirms that accumulating knowledge over long horizons is better than constantly starting from scratch. Instead, this ablation should isolate the contribution of the suggested fetch-top-K buffer (for example, against a baseline that randomly selects equations).
>
> **Response:** Thank you for this excellent point. We agree this is a valuable comparison and have conducted the suggested experiment. We compare the performance of top-k fetch and random-k fetch as follows:
>
> | Model |Method |  $\text{Acc}_{0.01}$| $\text{Acc}_{0.001}$|
> |---|---|---|---|
> | GPT-OSS-120B | top-k |63.57 | 49.35 |
> | GPT-OSS-120B|random-k | 58.14 | 41.86|
> | Qwen3-Coder-480B |top-k| 49.09 |24.55 |
> | Qwen3-Coder-480B |random-k |26.36 | 13.70 |
> | GLM-4.5-Air |top-k | 48.32|25.06 |
> | GLM-4.5-Air |random-k |41.96 | 21.71|
>
> As shown, the random-k fetch baselines significantly lag behind the top-k baselines, suggesting the importance of incorporating previously explored top-performing equations.  **We have included the new experiment in the revised manuscript (Page 9, Lines 464-467).**
>
> > **Question 1:** Have the authors considered the use of broader tools for data analysis, such data visualization or graphical analysis? Do you think that expanding the framework's tool capabilities could be useful?
>
> **Response:**  This is a great question. Analysis via plotting is supported by our framework, as the agent can write code using libraries like Matplotlib, and the `data analyzer` tool executes this code and returns the resulting figure. The primary modification required to fully integrate this capability would be to shift the backbone from a text-only model to a multimodal model capable of processing figure inputs. In fact, in the early stages of our work, we observed the agent already attempting to write code that included Matplotlib to visualize the data. However, because we mainly focused on text-only models, we restricted this behavior through specific prompts (Page 24, Line 1287).
>
> Regarding its effectiveness, recent work [1] shows that even current state-of-the-art models like GPT-5 and Gemini 2.5 Pro still lack the ability to infer accurate equations from figures. However, this remains a very promising direction, and we are optimistic that it can become effective with the improved capability of future foundation models.
>
> Lastly, we want to emphasize that while providing the model with only two tools (data analysis and equation evaluation) may seem limited in number, this impression is inaccurate. The data analysis tool is inherently a code interpreter. Through this single tool, the agent can write and execute diverse code to analyze the data, effectively encompassing many potential individual analysis tools. This design maintains a balance between universality (by allowing diverse code execution) and simplicity (by having a minimal set of defined tools).
>
> **We have revised the Tool Design section to make this point clear in the revised manuscript (Page 4, Lines 165-170). We have also added a Limitations and Future Directions section to discuss plot analysis (Page 10, Lines 503-513).**
>
> > **Question 2:** Could the authors please elaborate on the mechanisms that allow the framework to obtain a long-horizon behavior?
>
> **Response:** Thank you for the opportunity to elaborate on this. The long-horizon behavior is controlled by two hyperparameters: the number of iterations ($N$) and the maximum turns for each iteration ($M$). Taking $N=3$ as an example:
> - Iteration 1: The agent interacts with the environment until it reaches the maximum turns ($M$). Then, we add all explored equations from this iteration to the experience buffer.
> - Iteration 2: The agent clears the historical context from the previous iteration. It then fetches the top-k equations from the experience buffer to use as in-context examples and begins a new exploration phase until it again reaches the maximum turns ($M$). We then add all equations explored during this iteration to the experience buffer.
> - Iteration 3: The process repeats, identical to Iteration 2.
>
> The periodic clearing of historical information in the context window, along with the memory module implemented by the experience buffer, ensures the agent completes long-horizon exploration with a limited context window size.
>
> ---
> References:
>
> [1] Mimicking the Physicist's Eye: A VLM-centric Approach for Physics Formula Discovery, in arxiv 2025
>
> [2] Stress-strain data for aluminum 6061-t651 from 9 lots at 6 temperatures under uniaxial and plane strain tension, in Data in Brief 2019
>
> [3] LLM-SR: Scientific Equation Discovery via Programming with Large Language Models, in ICLR 2025

---

> ### Author Response · Authors · 2025-11-25
> **Looking forward to discussion**
>
> Dear Reviewer ewF1,
>
> We hope our answers and new experiments have addressed your concerns and questions. Please let us know if you have any more questions before the end of the discussion period.
>
> Should there be no additional concerns, we kindly ask you to consider revising your score.
>
> Thanks for your dedicated service to the community!

---

> > ### Comment · Reviewer_ewF1 · 2025-11-26
> >
> > My concerns about novelty and incremental work remain. However, your clarifications and additional experiments have addressed my other concerns related to ablations and real-world generalization. Therefore, I'm inclined to increase my score.

---

> > > ### Author Response · Authors · 2025-11-27
> > > **Official Comment by Authors**
> > >
> > > Dear Reviewer ewF1,
> > >
> > > Thank you for your continued engagement and for confirming that our clarifications and additional experiments have successfully addressed your concerns regarding ablations and real-world generalization.
> > >
> > > Regarding your remaining concern about novelty, and to clearly distinguish our work from previous LLM-based symbolic regression (SR) approaches, we have provided the comparison table below:
> > >
> > > | Method | Data Analysis | Flexibility | Training framework|
> > > |---|---|---|---|
> > > | LLM-SR [1] |✖️| ✖️ | ✖️|
> > > | LaSR [2] |✖️ | ✖️ | ✖️ |
> > > | SGA [3] |✖️ | ✖️ | ✖️|
> > > | Ours |✔️ |✔️| ✔️|
> > >
> > > *(Note: Detailed explanations can be found in our original response to Weakness 1.)*
> > >
> > > The aim of our work is to achieve a simple and general approach to scientific equation discovery that mimics the workflow of human scientists and demonstrates effective scalability with additional training. **We acknowledge that while agentic LLMs have been applied in domains such as software engineering and search, their integration into equation discovery remains underexplored.
> > > Moreover, from a technical standpoint, we wish to highlight that developing an agentic framework specifically for SR introduces novel components distinct from previous tasks:**
> > >
> > > - From an inference perspective: It requires carefully designing memory mechanisms capable of handling long-horizon continuous optimization.
> > >
> > > - From a training perspective: It necessitates the design of reward functions tailored for long-horizon continuous optimization. This differs significantly from the discrete feedback (e.g., pass/fail unit tests) commonly found in widely studied domains like mathematics or code generation.
> > >
> > > Furthermore, these long-horizon continuous optimization scenarios are not limited to SR but extend to other areas of scientific discovery, such as algorithm optimization. Thus, our methodology and analysis offer broader insights and benefits to these related fields.
> > >
> > > We are glad to hear you are inclined to increase your score, and we hope this clarification helps solidify that decision. We remain available if you need any final details before the discussion closes.
> > >
> > >
> > > Best regards,
> > >
> > > The Authors
> > >
> > >
> > > ---
> > > References:
> > >
> > > [1] LLM-SR: Scientific Equation Discovery via Programming with Large Language Models, in ICLR 2025
> > >
> > > [2] Symbolic Regression with a Learned Concept Library, in NeurIPS 2024
> > >
> > > [3] LLM and Simulation as Bilevel Optimizers: A New Paradigm to Advance Physical Scientific Discovery, in ICML 2024

---

### Official Review · Reviewer_E5Ep · 2025-11-02

**Soundness:** 3
**Presentation:** 3
**Contribution:** 3
**Rating:** 4
**Confidence:** 4

**Summary:**

The paper proposes SR-Scientist, an agentic large-language-model (LLM) framework for symbolic regression. It combines two tools (a data analyzer and an equation evaluator) in a long-horizon reasoning loop with an experience buffer and a reinforcement-learning (RL) fine-tuning stage based on GRPO. Experiments on the LSR-Synth benchmark across four scientific domains show solid empirical gains over existing SR and LLM baselines, supported by ablations and robustness analyses.

**Strengths:**

+ The paper is well motivated and clearly framed, presenting the idea of turning LLMs into autonomous “scientists” that iteratively refine hypotheses through data analysis and reasoning.
+ The evaluation covers multiple domains, five different LLM backbones, and includes analyses for noise robustness, out-of-domain generalization, and symbolic accuracy.
+ The modular two-tool framework with memory buffering is versatile and adaptable to other discovery tasks beyond symbolic regression.
+ The empirical results are consistently strong across settings and supported by detailed ablation studies.
+ The framework provides a practical system contribution that can serve as a foundation for other research works in scientific AI agents.

**Weaknesses:**

- The work mainly integrates existing components such as tool use, GRPO-based RL, and memory buffering, without introducing new learning principles. The proposed mechanisms are largely heuristic and not deeply analyzed.
- The RL fine-tuning stage performs worse than inference-only variants and is limited to a single backbone. Its “one-iteration” setup contradicts the long-horizon reasoning concept and lacks convergence or stability analysis.
- The comparison with baselines is not compute-normalized. SR-Scientist executes up to 25 turns × 40 iterations, while non-LLM baselines are capped at 100k equations, leading to possible resource imbalance.
- The metric choice emphasizes Accτ with a 5% trimming rule and omits standard regression metrics like RMSE or MAE. Results are averaged over only three runs, which weakens statistical confidence.
- Symbolic accuracy remains around 7%–8%, much lower than numeric accuracy (>60%), suggesting improved curve fitting but not genuine symbolic recovery.
- The paper lacks qualitative examples illustrating how the agent refines equations through multi-step reasoning, making the “scientific inspiration” claim speculative.
- Several parameters and mechanisms, such as K in the experience buffer, the MAPE goal threshold, and the exclusive use of BFGS for optimization, are presented without sensitivity analysis or justification.
- The dataset evaluation omits LSR-Transform due to contamination concerns, but does not demonstrate that LSR-Synth is free of memorization. There are no experiments on real-world scientific data.
- Figures are dense, tables inconsistent, and citations incomplete, which affects readability.
- In summary, the lack of algorithmic novelty, limited RL justification, resource imbalance, and weak interpretability analysis reduce the overall impact. The current version of the paper stands as a well-executed system but needs stronger methodological contributions, fairer comparisons, and deeper reasoning analysis to meet the bar for ICLR.

**Questions:**

- Why does RL underperform? Was training unstable or reward scaling flawed?
- How is data contamination avoided in LSR-Synth?
- Try to do an apples-to-apples comparison. Would SR-Scientist still outperform if all methods were given the same compute budget or wall-clock time?
- Why restrict the framework to two tools instead of adding others, like dimensional checks or simplification modules?
- What causes the observed drop in performance beyond 25 turns?
- Can you provide qualitative examples of reasoning trajectories that refine equations step by step?

---

> ### Author Response · Authors · 2025-11-20
> **Response to Reviewer E5Ep (1/4)**
>
> Thank you for your thoughtful review and constructive feedback. We appreciate your positive comments on our method's motivation, the strong performance, the detailed ablation studies, and the contribution to the broader AI community. We have provided a point-by-point reply to address your concerns as follows:
>
> > **Weakness 2:** The RL fine-tuning stage performs worse than inference-only variants and is limited to a single backbone. Its “one-iteration” setup contradicts the long-horizon reasoning concept and lacks convergence or stability analysis.
>
> > **Question 1:** Why does RL underperform? Was training unstable or reward scaling flawed?
>
> **Response:** Thank you for your feedback. We would like to clarify three key points:
> - Performance: The appropriate baseline for the RL fine-tuned model should be based on the same LLM backbone. For Qwen3-Coder-30B, RL fine-tuning yields a significant improvement by +8.62 compared to the non-fine-tuned version in Table 1. Furthermore, it outperforms other SR methods using the same backbone by +11.46, demonstrating strong performance.
> - Backbone Limitations: Regarding the concern that RL training is limited to a single backbone, we have provided results for RL training on three additional backbones in the appendix (Page 17, Lines 911-917) and referenced this in the main text (Page 6, Lines 314-315; Page 7, Lines 371-372).
> - "One-Iteration" Setup: The “one-iteration” setup does not contradict the concept of long-horizon reasoning. A single RL "iteration" is not equivalent to a single "interaction" with the environment. Within one iteration, our agent interacts with the environment for 20 turns with a 10K response length, which constitutes a substantial long-horizon exploration process.
>
> > **Weakness 1:**  The work mainly integrates existing components such as tool use, GRPO-based RL, and memory buffering, without introducing new learning principles. The proposed mechanisms are largely heuristic and not deeply analyzed.
>
> **Response:**  Thank you for this comment. Regarding the agentic frameworks or the tool use mentioned, we wish to highlight that directly adapting existing agentic models to the SR field is non-trivial, as it requires a specialized memory mechanism tailored for long-horizon continuous optimization. Unlike the memory buffering implemented by the non-agentic framework [2], it requires designing mechanisms to periodically clear historical information from the context window and replace it with summarized content.
>
> Similarly, our RL component goes beyond standard practices; it involves designing reward functions specifically for long-horizon continuous optimization scenarios, which differ from the discrete feedback (e.g., pass/fail tests) found in widely studied domains such as mathematics or coding.
>
> Moreover, we believe our extensive ablation studies and analysis in Section 4.3 address your concern that "the proposed mechanisms are largely heuristic and not deeply analyzed."
>
> > **Weakness 3:** The comparison with baselines is not compute-normalized. SR-Scientist executes up to 25 turns × 40 iterations, while non-LLM baselines are capped at 100k equations, leading to possible resource imbalance.
>
> > **Question 3:** Try to do an apples-to-apples comparison. Would SR-Scientist still outperform if all methods were given the same compute budget or wall-clock time?
>
> **Response:** Thanks for your feedback. Given the foundational differences between LLM-based methods and non-LLM baselines, a direct control of computational budget is challenging [2]. Moreover, the variance in LLM API call rates makes it hard to control for the same wall-clock time [4].
>
> Instead, following previous methods [1], we normalize each method by the hypothesis budget, i.e., the number of candidate equations for each problem. For all LLM-based methods, we constrained them to 1,000 LLM calls, with at most 1,000 hypothesis generations. Unlike previous work that assigns the same number of hypotheses for all methods [1], we increased the hypothesis budget for non-LLM based methods from 1X to 100X compared to the LLM-based method, providing a very strong baseline.

---

> ### Author Response · Authors · 2025-11-20
> **Response to Reviewer E5Ep (2/4)**
>
> > **Weakness 4:** The metric choice emphasizes Accτ with a 5% trimming rule and omits standard regression metrics like RMSE or MAE. Results are averaged over only three runs, which weakens statistical confidence.
>
>
> **Response:** We appreciate this comment. Regarding metric selection, we have detailed the rationale for prioritizing $\text{Acc}_{\tau}$ over other standard metrics like NMSE in the main text (Page 5, Lines 257-258). Specifically, metrics such as RMSE, MAE, and NMSE are unbounded, rendering them highly sensitive to extreme values when averaging performance across diverse problems. This point is further supported by our quantitative analysis in the appendix (Page 18, Lines 958-963).
>
> Regarding the second point, using three runs follows the standard setup for this benchmark [1].
>
> > **Weakness 5:** Symbolic accuracy remains around 7%–8%, much lower than numeric accuracy (>60%), suggesting improved curve fitting but not genuine symbolic recovery.
>
>
>
> **Response:** Thanks for your feedback. Symbolic accuracy is calculated based on whether the derived equation exactly matches the ground truth equation, which is an extremely challenging metric. The low absolute performance is observed across all SR methods, and we note that our method achieves the best performance among them. Moreover, we evaluate performance on the out-of-distribution (OOD) set to test if the derived equations can generalize. The strong performance on the OOD set indicates a higher degree of symbolic recovery, as these two metrics are highly correlated [1].
>
> > **Weakness 6:** The paper lacks qualitative examples illustrating how the agent refines equations through multi-step reasoning, making the “scientific inspiration” claim speculative.
>
> > **Question 6:** Can you provide qualitative examples of reasoning trajectories that refine equations step by step?
>
>
> **Response:** Thank you for this valuable suggestion. **We have included two qualitative examples of the agent's reasoning trajectories in the revised manuscript (Page 19, Lines 1021–1025).**
>
> > **Weakness 7:** Several parameters and mechanisms, such as K in the experience buffer, the MAPE goal threshold, and the exclusive use of BFGS for optimization, are presented without sensitivity analysis or justification.
>
> **Response:** Thank you for your valuable suggestions. To address the concerns regarding parameter sensitivity, we conducted additional analyses on the number of equations fetched from the experience buffer and the initial MAPE goal.
>
> Table 1: Sensitivity analysis of the number of equations ($K$) fetched from the experience buffer.
> | $K$ | $\text{Acc}_{0.01}$| $\text{Acc}_{0.001}$|
> |---|---|---|
> | 1 | 63.57 | 49.61 |
> | 3 |  63.57 | 49.35|
> | 5 | 63.31|50.90 |
> | 10 | 61.50|47.29 |
>
> Table 2: Sensitivity analysis of the initial MAPE goal ($s_{\text{goal}}$).
> | $s_{\text{goal}}$ | $\text{Acc}_{0.01}$| $\text{Acc}_{0.001}$|
> |---|---|---|
> | 0.1 | 64.34 | 48.84|
> | 0.01 |  63.31 | 49.61|
> | 0.001 |  63.57 | 49.35|
> | 0.0001 | 51.74|37.70|
>
>
> Regarding the number of equations fetched from the experience buffer, our method remains robust when this value is relatively small. However, performance declines slightly when it is increased to 10. This suggests that including a large number of fetched equations in the context window may strain the model’s capability.
>
> Regarding the initial MAPE goal, the results indicate that the performance is insensitive to relatively easy targets, as our dynamic goal update mechanism effectively prevents trivial goals. However, an extremely challenging goal (e.g., 0.0001) leads to degraded performance, causing the model to become trapped during the optimization process.
>
>
> Regarding the exclusive use of BFGS, we chose it over methods like Adam as it is preferred for problems with few parameters [2], fitting our scenario. Moreover, BFGS is also widely adopted in our baselines [1-3]. Adopting the same optimization algorithm ensures a fair comparison with them.
>
> **We have added these sensitivity analyses to the revised manuscript (Page 20, Lines 1028–1038).**

---

> ### Author Response · Authors · 2025-11-20
> **Response to Reviewer E5Ep (3/4)**
>
> > **Weakness 8:** The dataset evaluation omits LSR-Transform due to contamination concerns, but does not demonstrate that LSR-Synth is free of memorization. There are no experiments on real-world scientific data.
>
> > **Question 2:** How is data contamination avoided in LSR-Synth?
>
> **Response:** Thanks for your feedback. Regarding your first point on memorization concerns, we would like to clarify the distinction between the curation of LSR-Transform and LSR-Synth:
>
> - LSR-Transform is based on transforming known equations by switching the roles of a selected input feature and the original target variable. Consequently, the underlying equation structure is inherently known.
> - LSR-Synth, in contrast, synthesizes novel terms to create equations distinct from those found in existing literature. These generated equations are designed to prevent memorization while being validated by subject matter experts to ensure scientific rigor.
>
> Regarding your second point that "There are no experiments on real-world scientific data," we wish to highlight that the LSR-Synth dataset is grounded in realistic scientific contexts through the following measures:
>
> - All equations, along with visualizations of their generated data points, were cross-checked by two subject matter experts to validate their scientific rigor.
> - All constants and variable ranges within the equations are designed to fit plausible physical scenarios.
> - Each equation is composed of both known and synthetic terms. Known terms refer to those commonly found in scientific equations and are well-documented in the literature for that domain and specific problem. These terms are included to ensure that the problems remain grounded in the established scientific context.
>
> Moreover, we have conducted additional experiments on the Stress-Strain dataset [5] to address your concern. This benchmark problem leverages a real-world experimental dataset, comprising tensile tests on Aluminum 6061-T651 across a range of temperatures. The inclusion of this benchmark helps test the performance on real-world experimental data. Moreover, there is no predetermined theoretical model structure for this problem, thereby avoiding memorization issues. The results are as follows:
>
> Table 3: Performance on the Stress-Strain dataset.
> | Method | $\text{Acc}_{0.1}$| $\text{Acc}_{0.01}$|
> |---|---| ---|
> | PySR | 70.67 | 7.91 |
> | LaSR |  63.80 | 7.21 |
> | LLM-SR | 68.52| 8.74 |
> | SR-Scientist | 76.98  |16.02 |
>
> As shown, SR-Scientist outperforms other methods, demonstrating its effectiveness on real-world data. **We have added the clarification regarding the LSR-Synth dataset (Page 5, Lines 250-252) and included the new experiment (Page 8, Lines 429-431) in the revised manuscript.**
>
> > **Weakness 9:** Figures are dense, tables inconsistent, and citations incomplete, which affects readability.
>
> **Response:** Thank you for pointing this out. **We have addressed the issue of dense figures by utilizing the additional page allowed in the rebuttal phase to improve layout and clarity.** Regarding the table inconsistencies and incomplete citations, could you please provide specific examples? We would be happy to correct them immediately.
>
> > **Question 4:** Why restrict the framework to two tools instead of adding others, like dimensional checks or simplification modules?
>
> **Response:** That is an excellent question. Our framework's two tools do not restrict these functions. The agent can perform dimensional checks or simplification by writing code and the `data analyzer` tool executes it and returns the results. While providing the model with only two tools (data analysis and equation evaluation) may seem limited in number, this impression is inaccurate. The `data analyzer` tool is inherently a code interpreter. Through this single tool, the agent can write and execute diverse code to analyze the data, effectively encompassing many potential individual analysis tools. This design maintains a balance between universality (by allowing diverse code execution) and simplicity (by having a minimal set of defined tools).
>
> **We have revised the Tool Design section to make this point clear in the revised manuscript (Page 4, Lines 165-170).**

---

> ### Author Response · Authors · 2025-11-20
> **Response to Reviewer E5Ep (4/4)**
>
> > **Question 5:** What causes the observed drop in performance beyond 25 turns?
>
> **Response:** Thank you for this insightful question. First, we would like to first clarify our experimental setup and objectives. In our experiments, we maintain a fixed budget of 1,000 LLM calls, which requires a trade-off between the number of iterations and the maximum number of turns allowed per iteration. For example, a limit of 25 maximum turns allows for 40 iterations, whereas increasing the limit to 30 turns reduces the budget to approximately 33 iterations. This setup helps us analyze the benefits of increasing the interaction depth (turns) per iteration at the cost of reducing the total number of iterations.
>
> Regarding the results, increasing the maximum turns from 10 to 25 yields significant performance improvements, highlighting the value of long-horizon optimization. However, the stagnation or slight decline observed beyond 25 turns does not imply that 30 turns are less effective than 25 turns given the same number of iterations. Rather, it indicates that at this point, the fixed inference budget is better allocated to initiating new iterations. We attribute this to the fact that a larger number of turns (e.g., 30) challenges the model's long-context capabilities, causing it to become trapped in local exploration strategies despite the increased interaction depth.
>
> ---
> References:
>
> [1] LLM-SRBench: A New Benchmark for Scientific Equation Discovery with Large Language Models, in ICML 2025
>
> [2] LLM-SR: Scientific Equation Discovery via Programming with Large Language Models, in ICLR 2025
>
> [3] Neural Symbolic Regression that Scales, in ICML 2021
>
> [4] Symbolic Regression with a Learned Concept Library, in NeurIPS 2024
>
> [5] Stress-strain data for aluminum 6061-t651 from 9 lots at 6 temperatures under uniaxial and plane strain tension, in Data in Brief 2019

---

> ### Author Response · Authors · 2025-11-25
> **Looking forward to discussion**
>
> Dear Reviewer E5Ep,
>
> We hope our answers and new experiments have addressed your concerns and questions. Please let us know if you have any more questions before the end of the discussion period.
>
> Should there be no additional concerns, we kindly ask you to consider revising your score.
>
> Thanks for your dedicated service to the community!

---

> > ### Comment · Reviewer_E5Ep · 2025-11-27
> >
> > Thank you to the authors for the detailed and thoughtful rebuttal. I appreciate the substantial effort in addressing the concerns. I think these additions improve the overall clarity and completeness of the submission. I increased my score. I believe the paper is appropriate for presentation at the conference, provided that the authors incorporate the clarifications, additional experiments, and explanations from the rebuttal into the camera-ready. Strengthening the main text with these details will significantly improve accessibility and long-term impact.

---

> > > ### Author Response · Authors · 2025-11-27
> > > **Thank you from Authors**
> > >
> > > Thank you once again for your time and valuable insights. We are glad that our rebuttal has resolved your concerns and appreciate the raised score.

---

### Official Review · Reviewer_FL8G · 2025-11-02

**Soundness:** 4
**Presentation:** 3
**Contribution:** 4
**Rating:** 8
**Confidence:** 5

**Summary:**

This paper presents a novel framework, SR-Scientist, for LLM-based scientific equation discovery. Unlike prior approaches that primarily treat LLMs only as equation proposers within search algorithms, SR-Scientist extends their role to act as agentic AI scientists capable of performing complementary pre- and post-processing tasks such as data analysis, equation implementation, and submission for evaluation. The framework integrates a set of wrapped tools (e.g., data analysis and equation evaluation) that are used in an agentic and long-horizon manner to autonomously navigate the full equation discovery process.
Empirical results demonstrate that the proposed approach outperforms existing baselines on standard benchmarks, exhibits improved robustness to noise, and shows better generalization across diverse tasks. The paper also includes experiments with a smaller Qwen-30B model augmented with end-to-end RL fine-tuning which shows their results could even be strengthen with the fine-tuning and adaptation of LLM backbone.

**Strengths:**

- The paper is well-written and clearly motivated, making it an engaging read.
- The experiments are comprehensive on recent benchmarks and with thorough analysis.
- The reported results demonstrate substantial performance improvements over state-of-the-art baselines.

**Weaknesses:**

- The proposed data analysis tool is well-motivated and conceptually sound. However, I am unclear about the necessity of using an LLM as an equation evaluator, given that evaluation in this task is typically data-driven and follows a consistent procedure. Introducing an agentic LLM into the evaluation loop could potentially introduce unnecessary variability or LLM generation errors. I would appreciate clarification on why this component is needed and what advantages it offers over standard predefined evaluation methods for generated hypotheses. Additionally, Table 3 seem to not include an ablation for this tool (T2), making its specific role and impact questioning.

- I think additional qualitative examples would help clarify what the agentic steps contribute beyond the baseline LLM-SR approach. Showing intermediate output such as snippets of generated code or key decision points the agent follows during the discovery process would help to understand how these steps influence the final discovered equations.

- For the results reported in Table 2 and Figure 3, it is unclear whether they are evaluated on all LLM-SRBench tasks or only a specific subset or category. It would be helpful to include this information explicitly in the captions.

**Questions:**

Included in the weaknesses section

---

> ### Author Response · Authors · 2025-11-20
> **Response to Reviewer FL8G**
>
> Thank you for your thoughtful review and constructive feedback. We appreciate your positive comments on our method's novelty, strong performance, and detailed analysis. We have provided a point-by-point reply to address your concerns below.
>
> > **Weakness 1:** The proposed data analysis tool is well-motivated and conceptually sound. However, I am unclear about the necessity of using an LLM as an equation evaluator, given that evaluation in this task is typically data-driven and follows a consistent procedure. Introducing an agentic LLM into the evaluation loop could potentially introduce unnecessary variability or LLM generation errors. I would appreciate clarification on why this component is needed and what advantages it offers over standard predefined evaluation methods for generated hypotheses. Additionally, Table 3 seem to not include an ablation for this tool (T2), making its specific role and impact questioning.
>
> **Response:** Thank you for your insightful comment. We would like to clarify that the LLM does not act as the equation evaluator itself. Instead, the `equation evaluator` is a deterministic, program-based tool. The LLM functions as an agent that invokes this tool by passing a generated equation; the tool then executes standard code to calculate performance metrics following a consistent procedure. This design ensures there is no LLM-induced variability or hallucination in the evaluation step. **We have carefully revised the description in the Tool Design section to resolve any ambiguity in the updated manuscript (Page 4, Lines 162–163).**
>
> > **Weakness 2:** I think additional qualitative examples would help clarify what the agentic steps contribute beyond the baseline LLM-SR approach. Showing intermediate output such as snippets of generated code or key decision points the agent follows during the discovery process would help to understand how these steps influence the final discovered equations.
>
> **Response:** Thank you for your suggestion. **We have included two qualitative examples demonstrating how the agent derives insights from data analysis and long-horizon optimization in the revised manuscript (Page 19, Lines 1021–1025).**
>
> > **Weakness 3:** For the results reported in Table 2 and Figure 3, it is unclear whether they are evaluated on all LLM-SRBench tasks or only a specific subset or category. It would be helpful to include this information explicitly in the captions.
>
> **Response:** We appreciate your valuable suggestion. Table 2 and Figure 3 report results on the full LSR-Synth dataset. **We have revised the captions to explicitly clarify this in the updated manuscript (Page 7, Line 357).**

---

> > ### Comment · Reviewer_FL8G · 2025-11-20
> > **Official Comment by Reviewer FL8G**
> >
> > Thanks for clarifying. I'm still unclear on how your evaluation procedure differs from LLM-SR. If you're just always calling an external evaluation program to execute extracted equation, why frame this as a separate tool? I see two possible scenarios in this case: (1) The LLM decides when to invoke the evaluator (deciding when and where to do tool calling); (2) The LLM always calls the evaluator automatically (no decision-making). In my opinion, scenario 2 is identical to LLM-SR's approach which is just execution without the "tool" label, and scenario 1 introduces unnecessary risk in the evaluation procedure: the LLM might hallucinate about when to call the tool which is not needed in this problem, adding unwanted variation to what should be straightforward program execution.
> > I think that tools are meaningful when deciding whether and when to use them matters. For the data analysis part, this makes sense to me. However, for simple evaluator program execution, that doesn't seem to apply. Can you clarify what motivates treating evaluation as a tool-calling in your approach to this problem?

---

> ### Author Response · Authors · 2025-11-21
> **Response to the follow-up questions from Reviewer FL8G**
>
> Thank you for the follow-up and the detailed description; this is an excellent question.
>
> To ensure we fully address your concern, we interpret your question as follows: *Given that equation evaluation is a deterministic step, why use an LLM-driven "tool call" format (which implies a risk of decision-making variance) rather than a hard-coded execution loop like LLM-SR?*
>
> **In summary, while we agree that equation evaluation functions as a deterministic execution step, we implement it via the "tool call" format to unify the agent's interaction interface across different tasks (data analysis vs. equation proposal) and to ensure robust parsing infrastructure.**
>
> Our detailed explanation is as follows:
>
> 1. Unified Multi-Turn Interaction Interface
>
> Unlike LLM-SR, which typically focuses on a single task pipeline, our agent performs mixed tasks: it conducts data analysis and equation proposal. By treating "Equation Evaluation" as a tool, we unify the agent's interaction interface. The model interacts with the environment exclusively through a standardized tool-calling protocol for both exploratory data analysis and equation submission. This facilitates a much more streamlined implementation of the inference and training infrastructure.
>
> 2. Infrastructure and Robustness
>
> From an engineering perspective, the tool-call format significantly simplifies the control flow in multi-turn interaction with the environment.  During interaction, the inference infrastructure must handle the following steps:
>
> - Deciding when to stop decoding based on predefined stopping tokens (usually the *eos* token).
>
> - Parsing the decoded text (e.g., the equation) and passing the content to the external environment (e.g., the code interpreter).
>
> - Concatenating the feedback (e.g., the equation's performance) from the environment to the context  and continuing generation.
>
> Crucial to this process is the stopping token for decoding. Models post-trained for agentic tasks automatically generate the *eos* token at the end of a tool call; thus, the tool-call format inherently achieves the necessary flow control. While it is possible to avoid the tool format, for example, by instructing the model to output specific tags at the end of an equation and adding them as custom stop words in inference engines like vLLM, this is less reliable. Models can fail to output predefined custom special tokens because they are not explicitly trained on them, whereas they are highly optimized for standard tool-calling formats.
>
> 3. Empirical Analysis
>
> Finally, to address your concern that the agent might "hallucinate about when to call the tool" (Scenario 1), we empirically analyzed the model's behavior. We classified the generated equations into two categories:
> - **Direct Tool Invocation:** The equation is generated directly as an argument within the `equation evaluation` tool call.
> - **Text-First Generation:** The equation is generated in the main text, requiring a subsequent "decision" step to invoke the tool.
>
> The percentage of equations generated directly within the tool call is reported below:
>
> | Model | Percentage (%)|
> |---|---|
> | GPT-OSS-120B | 99.65 |
> | Qwen3-Coder-480B | 99.96 |
> | Qwen3-Coder-30B | 99.60 |
> | GLM-4.5-Air | 99.80 |
>
>
> As shown, >99.5% of the proposed equations are immediately wrapped in the automatic evaluation tool call. This confirms that our setup functions empirically as "Scenario 2" (automatic execution), successfully avoiding the variability risks you mentioned while retaining the architectural benefits of a unified tool interface.
>
>
> We hope this clarification fully addresses your concerns.

---

> > ### Author Response · Authors · 2025-11-25
> > **Looking forward to discussion**
> >
> > Dear Reviewer FL8G,
> >
> > We hope our reply has addressed your concerns and questions. Please let us know if you have any more questions before the end of the discussion period.
> >
> > Thanks for your dedicated service to the community！

---

### Official Review · Reviewer_zbXv · 2025-11-04

**Soundness:** 3
**Presentation:** 3
**Contribution:** 3
**Rating:** 8
**Confidence:** 4

**Summary:**

This paper introduces SR-SCIENTIST, a framework for applying large language models (LLMs) to the scientific equation discovery (SED) process. The framework integrates evolutionary search with an LLM-as-heuristic component but offers greater flexibility than prior methods by allowing LLMs to perform a wider range of actions in each iteration. The paper demonstrates that enabling LLMs to interact directly with observational data (via tool use) and to engage in multi-turn interaction improves performance on SED tasks. Additionally, it explores finetuning LLMs with GRPO for this domain.

**Strengths:**

1. The experiments present strong results across multiple metrics and settings, consistently favoring the proposed approach.
2. Comprehensive ablation studies highlight the effects of two key components - tool calls and experience buffer - as well as the influence of interaction length and the distribution of tool usage.
3. The paper provides a detailed discussion of the reinforcement learning (RL) process, including the methodology for constructing synthetic training data and a comparative analysis of different reward function designs.
4. The writing is clear, well-structured, and easy to follow.

**Weaknesses:**

1. The contribution in terms of novelty is somewhat incremental. The framework closely resembles prior approaches such as LLM-SR, particularly in its use of an experience buffer, evaluation module, and parameter optimization function. The main components - tool use and multi-turn interaction - can be seen as adaptations of existing agentic LLM paradigms (e.g., GPT-OSS, Qwen3-Coder) to the symbolic regression task, rather than a fundamentally new agentic framework. The RL fine-tuning pipeline largely follows established practices, including synthetic data construction and the use of the GRPO loss.

**Questions:**

1. Figure 6 shows that Equation Evaluation dominates tool usage. Does this suggest that the agents primarily engage in a “guess-and-check” process rather than deriving insights from observation data through the Data Analyzer tool?
2. The analysis indicates that increasing the maximum number of turns beyond 25 results in stagnating or slightly declining performance. Is this due to context length limitations, or does it reflect the agent’s inability to explore new strategies (e.g., requiring periodic resetting or reinitialization)?
3. Could the authors include representative failure cases? A qualitative analysis of typical failure modes - such as difficulties with specific mathematical structures, high noise levels, or certain data domains - would provide valuable insights into the framework’s limitations.

---

> ### Author Response · Authors · 2025-11-20
> **Response to Reviewer zbXv (1/2)**
>
> Thank you for your thoughtful review and constructive feedback. We appreciate your positive comments on our method's flexibility, strong performance, and detailed analysis. We have provided a point-by-point reply to address your concerns below.
>
> > **Weakness 1:** The contribution in terms of novelty is somewhat incremental. The framework closely resembles prior approaches such as LLM-SR, particularly in its use of an experience buffer, evaluation module, and parameter optimization function. The main components - tool use and multi-turn interaction - can be seen as adaptations of existing agentic LLM paradigms (e.g., GPT-OSS, Qwen3-Coder) to the symbolic regression task, rather than a fundamentally new agentic framework. The RL fine-tuning pipeline largely follows established practices, including synthetic data construction and the use of the GRPO loss.
>
> **Response:** Thank you for this comment. To clearly distinguish our work from previous LLM-based symbolic regression (SR) approaches, we have provided the comparison table below:
>
> | Method | Data Analysis | Flexibility | Training framework|
> |---|---|---|---|
> | LLM-SR [1] |✖️| ✖️ | ✖️|
> | LaSR [2] |✖️ | ✖️ | ✖️ |
> | SGA [3] |✖️ | ✖️ | ✖️|
> | Ours |✔️ |✔️| ✔️|
>
> Key distinctions are detailed below:
>
> - Integration of Data Analysis Tools. Existing LLM-based SR methods lack direct analysis of observed data through tools to gain insights. By introducing data analysis tools, our method shifts the paradigm from enumerative search to data-driven reasoning and analysis. This is particularly critical for discovering novel equations where prior knowledge is limited.
>
> - Flexible Agentic Framework. Unlike previous frameworks that rely on fixed workflows, our agentic framework empowers the LLM to dynamically determine its next action based on environmental feedback. **While agentic LLMs exist in fields like software engineering and search, their integration into equation discovery remains underexplored.**
>
> - Comprehensive Training Pipeline.  Most existing LLM-based SR approaches focus solely on inference, overlooking the potential of task-specific training. We provide a comprehensive pipeline, ranging from training data construction to reward design, enabling the LLM to enhance its capabilities through RL.
>
> **From a technical standpoint, we wish to highlight that developing an agentic framework specifically for SR introduces novel components distinct from previous tasks:**
>
> - From an inference perspective: It requires carefully designing memory mechanisms capable of handling long-horizon continuous optimization.
>
> - From a training perspective: It necessitates the design of reward functions tailored for long-horizon continuous optimization. This differs significantly from the discrete feedback (e.g., pass/fail unit tests) commonly found in widely studied domains like mathematics or code generation.
>
> Furthermore, **these long-horizon continuous optimization scenarios are not limited to SR but extend to other areas of scientific discovery, such as algorithm optimization.** Thus, our methodology and analysis offer broader insights and benefits to these related fields.
>
>
> > **Question 1:** Figure 6 shows that Equation Evaluation dominates tool usage. Does this suggest that the agents primarily engage in a “guess-and-check” process rather than deriving insights from observation data through the Data Analyzer tool?
>
> **Response:** Thank you for this excellent question. We observe that tool usage behavior is highly model-dependent. As shown in Figure 6, GPT-OSS-120B engages in data analysis significantly more frequently (64%) compared to other models. GPT-OSS-120B also demonstrates superior performance in Table 1, suggesting a positive correlation between data analysis and performance.
> Furthermore, our ablation studies (Table 4) show that removing the data analysis tool leads to decreased performance, confirming its necessity.
>
> **To provide direct evidence that the agent derives insights from observation data via the data analysis tool, we have added two qualitative case studies in the revised manuscript (Page 19, Lines 1021–1025).** We appreciate you highlighting this point.
>
> ---
> References:
>
> [1] LLM-SR: Scientific Equation Discovery via Programming with Large Language Models, in ICLR 2025
>
> [2] Symbolic Regression with a Learned Concept Library, in NeurIPS 2024
>
> [3] LLM and Simulation as Bilevel Optimizers: A New Paradigm to Advance Physical Scientific Discovery, in ICML 2024

---

> > ### Author Response · Authors · 2025-11-20
> > **Response to Reviewer zbXv (2/2)**
> >
> > > **Question 2:** The analysis indicates that increasing the maximum number of turns beyond 25 results in stagnating or slightly declining performance. Is this due to context length limitations, or does it reflect the agent’s inability to explore new strategies (e.g., requiring periodic resetting or reinitialization)?
> >
> > **Response:** Thank you for this insightful question. First, we would like to  clarify our experimental setup and objectives. In our experiments, we maintain a fixed budget of 1,000 LLM calls, which requires a trade-off between the number of iterations and the maximum number of turns allowed per iteration. For example, a limit of 25 maximum turns allows for 40 iterations, whereas increasing the limit to 30 turns reduces the budget to approximately 33 iterations. This setup helps us analyze the benefits of increasing the interaction depth (turns) per iteration at the cost of reducing the total number of iterations.
> >
> > Regarding the results, increasing the maximum turns from 10 to 25 yields significant performance improvements, highlighting the value of long-horizon optimization. However, the stagnation or slight decline observed beyond 25 turns does not imply that 30 turns are less effective than 25 turns given the same number of iterations. Rather, it indicates that at this point, the fixed inference budget is better allocated to initiating new iterations. We attribute this to the fact that a larger number of turns (e.g., 30) challenges the model's long-context capabilities, causing it to become trapped in local exploration strategies despite the increased interaction depth.
> >
> > > **Question 3:** Could the authors include representative failure cases? A qualitative analysis of typical failure modes - such as difficulties with specific mathematical structures, high noise levels, or certain data domains - would provide valuable insights into the framework’s limitations.
> >
> > **Response:** Thank you for this valuable suggestion. **We have added a Limitations and Future Directions section in the revised manuscript to discuss failure cases based on our qualitative analysis (Page 10, Lines 503–513).**

---

> ### Author Response · Authors · 2025-11-25
> **Looking forward to discussion**
>
> Dear Reviewer zbXv,
>
> We hope our answers and new experiments have addressed your concerns and questions. Please let us know if you have any more questions before the end of the discussion period.
>
> Thanks for your dedicated service to the community！

---

### Meta-Review · Area_Chair_3YXL · 2025-12-30

**Summary:**

The biggest weakness from numerous reviews is the novelty of this work over LLM-SR, in that it just replaces LLM with a more agentic framework. Other weaknesses included things like sensitivity analysis, lack of qualitative results, and real-world examples. Moreover, some reviewers suggest a lack of limitations section.

**Reviewer Concerns:**

The authors have added new experiments thatt address  sensitivety analysis, added qualitative results and real-world usecases. The authors have also added discussion on limitations of the method.

However the problems with technical novelty is still not resolved. Reviewer ewF1 was not convinced till the end. Similarly, Invoking the use of an evaluator as a tool is not really an innovation as stated by FL8G. The reviewer were not convinced.

Nonetheless both the reviewers with a score of 4 are raising their scores, so the consensus is unanimously positive.

**Reviewer Scores:**

zbXv, FL8G: Scores will stay at 8

E5Ep, ewF1: stated they are inclined to increase the score. So it should be 6.

---

### Decision · Program_Chairs · 2026-01-26

Accept (Poster)